# The influence of hip muscle strength on gait in individuals with a unilateral transfemoral amputation

Daniel Walter Werner Heitzmann[1,2]*, Julien Leboucher[1], Julia Block[1,2], Michael Günther[3], Cornelia Putz[1,2], Marco Götze[1], Sebastian Immanuel Wolf[1], Merkur Alimusaj[1,2]

**1** Motion Analysis Lab, Department of Orthopaedics and Trauma Surgery, Heidelberg University Hospital, Heidelberg, Germany, **2** Department of Orthopaedics and Trauma Surgery, Prosthetics and Orthotics Department, Heidelberg University Hospital, Heidelberg, Germany, **3** Guenther Bionics GmbH, Magdeburg, Germany

* daniel.heitzmann@med.uni-heidelberg.de

## Abstract

**Data Availability Statement:** All relevant data are within the manuscript and its Supporting Information files.

## Introduction

A unilateral transfemoral amputation (TFA) has a major impact on function. A leg-length discrepancy is the primary structural change, accompanied by the loss of lower-limb muscle volume and function. Prostheses can help individuals with a TFA to regain function, but such individuals still do not reach the functional level of unimpaired peers and exhibit gait deviations. This study gives insight into the causality between residual limb strength and gait deviations in individuals with a TFA.

## Methods

A convenient sample of 13 male individuals with a TFA (38.0 ± 12.6y; 179.7cm ± 6.5cm; 82.9kg ± 12.4kg) was recruited for this study. One participant with TFA was excluded, as he differed from the rest of the cohort, in residual limb length and the use of walking aids. A cohort of 18 unimpaired subjects served as a reference group (REF; nine females; 44y ± 13y; 174cm ± 9cm; 71kg ± 12kg). All participants underwent a conventional clinical gait analysis using a marker based 3D motion capture system and force platforms. Kinematics and kinetics were determined utilizing standard modelling methods. All subjects underwent a strength test, using a custom-made device to determine isometric moments of the hip joint in abduction, adduction, extension, and flexion. Peak values for maximum isometric moments for each movement direction and selected kinematic and kinetic values were derived from the results. Differences between subjects with TFA and unimpaired were compared using a Mann-Whitney U Test and associations between groups by Spearman's rank correlation.

**Funding:** This study was supported by Bundesministerium für Arbeit und Soziales (BMAS) - Referat Orthopädische Hilfmittel in the form of a grant awarded to DWWH, JB, SIW and MA (Title 684 02) and Deutsche Forschungsgemeinschaft (DFG) in the form of a grant awarded to DWWH, JL, JB, CP, SIW, and MA within the project 'Biomechanik des Prothesengangs' (project no. 322886417). Guenther Bionics GmbH also provided support in the form of a salary for MG. The specific roles of this author are articulated in the 'author contributions' section. The funders had no role in study design, data collection and analysis, decision to publish, or preparation of the manuscript.

**Competing interests:** The authors have read the journal's policy and have the following competing interests: Guenther Bionics GmbH is owned by MG, who was involved in study design and data collection prior to establishing the company. This does not alter our adherence to PLOS ONE policies on sharing data and materials. There are no patents, products in development or marketed products associated with this research to declare.

**Abbreviations:** CGA, Conventional clinical gait analysis; MFCL, Medicare 5-level functional classification system; MIM, Maximum isometric joint moment; OpTiMo, Device for testing isometric moments (OpTIMo = Optical Testing Isometric Moment); REF, unimpaired subjects served as a reference group; TFA, transfemoral amputation; TTA, transtibial amputation.

## Results

The participants with a TFA showed a significantly lower maximum isometric moment for hip abduction (0.85 vs. 1.41 Nm/kg p < .001), adduction (0.87 vs. 1.37 Nm/kg p = .001) and flexion (0.93 vs. 1.63 Nm/kg p = .010) compared to the reference group. Typically reported gait deviations in people with a TFA were identified, i.e. significant lower cadence and increased step width. We further identified altered coronal plane hip and trunk kinematics, with significantly higher ranges of motion during involved side stance-phase. Gait kinetics of individuals with a TFA showed significantly lower peak values during stance for hip abduction, adduction and extension moments in comparison to the reference group. We identified a moderate negative correlation between maximum isometric moment for hip abduction and trunk obliquity range of motion (ρ = -0.45) for participants with a TFA, which was not significant (p = 0.14).

## Conclusion

We showed that there are strength deficits in individuals with TFA and, that there are moderate correlations between gait deviations, i.e. lateral trunk lean during involved side stance and isometric hip abductor moment. The relation between maximum moments during gait and the corresponding maximum isometric moment may therefore be helpful to detect strength related compensation mechanisms. However, the moderate, non-significant correlation between lateral trunk lean and isometric hip abductor moment was the only one which corresponded directly to a gait deviation. Thus results must be interpreted with care. This study suggests that gait deviations in individuals with TFA are multifactorial and cannot be exclusively explained by their strength deficits. Future studies should explore the relationship between strength with kinematics and kinetics during gait in this population.

## Introduction

Unilateral lower-limb amputation has a major impact on individuals' body integrity, perception and physical capabilities. A leg-length discrepancy between the involved and sound side represents the primary structural change, accompanied by the loss of functional levers, e.g., the foot lever. The amputation also has further impact on lower-limb muscle function. Muscle insertions distal to the amputation site are lost, making part of the lower limb musculature ineffective. Some further musculoskeletal comorbidities following lower-limb amputation are: low back pain, osteoarthritis, reduced bone density and volume loss and atrophy of involved muscles [1]. These changes related to muscle function lead to a deficit in muscle strength, as reported in previous studies investigating individuals with a transtibial amputation (TTA) [2–8] and transfemoral amputation (TFA) [5, 7, 9–12]. Although the impact on muscle function is obvious, it remains unclear how the deficit in strength relates to function and, more specifically, to walking. It has been shown that adequate strength is needed in both the residual and sound limb to improve walking ability in people with TTA [4]. Furthermore, it was also concluded in a review that the evidence of a relationship between muscle strength and walking ability remains insufficient in people with a lower limb amputation [13].

Via ultrasound imaging, involved side atrophy of the anterior thigh muscles was detected in individuals with TTA [6]. This led to altered walking biomechanics and residual limb muscle

activation patterns after a TTA [6]. Also in individuals with TFA, muscle atrophy of the hip abductors was identified using MRI imaging [14]. However, neither study associated the functional muscular changes with measurements of muscle strength. In addition to these specific changes, general differences of the muscles are prevalent following amputation at both levels, e.g. larger amounts of fat embedded in the residual limb. Such fatty muscle degeneration was found, when residual limb muscles were compared to the corresponding muscles of the uninvolved side [15, 16].

The prosthetic socket may also have an influence on residual limb strength. The interface formed by the residual limb and the socket provides a semi-articulate coupling between the residuum and the prosthesis, due to incongruent movement between socket and stump influenced by soft tissue and socket fit. This results in a pseudo joint, which will not transfer loads through the lower extremity as in an unimpaired person, possibly leading to altered muscle loads. So, residual limb muscles in those with TFA will not be exposed to the same level of exercise, as lower limb muscles in unimpaired persons. In contrast to this, a more rigid fixation of the residual limb with the prosthesis, i.e. by osseointegration, or bone anchored prostheses, in individuals with TFA can have a positive effect on muscle strength [17].

Several studies have shown a general increase in function in people with lower limb amputation when they exercise, and they therefore demonstrated an increase in lower limb strength. Isometric strength training led to increased residual limb volume, improved suspension of the prosthetic socket, and gave a mean increase in walking speed by 13% when comparing results before and after the intervention [18]. Similar effects were reported for an isokinetic strengthening program with some participants doubling their walking distance after the program [19]. Individuals with TTA who frequently participate in sports activities were stronger than their non-active peers, reaching strength magnitudes almost as large as an unimpaired control group, while inactive individuals with TTA were significantly weaker [20].

Besides general function, walking and gait deviations are often associated with lower limb strength deficits. Excessive coronal plane movement of the trunk is one well known example of a gait deviation which is induced by a strength deficit [21]. It is commonly attributed to weakness of the hip abductors. Indeed, there is a relationship between the hip abductor strength and irregularities in trunk movements in several different pathological conditions [22–24]. Such deviations have also been reported in individuals with TFA [25, 26]. To date, it has not been shown that coronal plane trunk deviations in those subjects with TFA are caused primarily by weak hip abductors as these deviations may also be related to other factors, such as socket related issues. Leg length discrepancy is another potential cause for trunk deviation in people with TFA. A prosthetic leg length discrepancy could be due to one or a number of factors: Pistoning of the residual limb within the socket, the compression of elastic prosthetic feet or intentionally shortened prosthetic legs to guarantee clearance in swing.

To determine how muscle strength and walking are related in individuals with TFA we measured isometric strength, utilizing a custom made testing rig [11]. Following the ideas of Fosang et al. and Dallmeijer et al., who used a similar approach in individuals with cerebral palsy, we wanted to compare the maximum isometric joint moments (MIM) of people with TFA with the corresponding joint moments during walking as determined by conventional clinical gait analysis (CGA) [27–29]. We developed the following hypotheses for this study: Firstly, that people with TFA would have lower isometric moments on the involved side than those in the reference group; secondly, we expected to find similar gait deviations in individuals with TFA, as previously described in the literature, i.e. deviations in trunk movement. Finally, we expected to find correlations between maximum isometric joint moments and parameters of gait kinematics and kinetics, in that we expected weaker participants with TFA would show greater deviations during walking.

## Material and methods

The parameters investigated here were the hip MIM, temporal spatial parameters, kinematics, and kinetics during walking. For MIM, at least two valid repeated measurements were collected for each of the four movement directions flexion, extension, abduction, and adduction. The trial with the highest MIM was used for subsequent analysis. From the CGA data, maximum hip moments as well as ranges of motion of trunk, pelvis, and hips in all three planes and temporal-spatial parameters (step length, step time, walking speed, cadence, and step width) were calculated. Group averages were analysed for the residual side only. For the reference group both limbs were used for calculating the group average. Methodological details are specified in the following paragraphs.

### Participants

This study was approved by the Institutional Review Board (Ethikkommission der Medizinischen Fakultät Heidelberg; reference number S-302/2007). Recruitment was conducted at the in-house prosthetics and orthotics department. Prior to the assessment, written informed consent was obtained from all participants. A total of 13 male participants with TFA (38.0 ± 12.6y; 179.7cm ± 6.5cm; 82.9kg ± 12.4kg; details in Table 1) completed CGA and subsequently underwent a strength assessment in which the maximum isometric hip joint moment for abduction, adduction, flexion, and extension of the hip on the involved side were measured. The OpTIMo device [11] was utilized for this assessment and will be described in a following section (Isometric muscle strength measurements OpTIMo).

**Table 1. Detailed information of the participants with a transfemoral amputation (TFA) and their current prostheses.**

| Person with TFA | cause for amputation | height [cm] | mass* [kg] | age [y] | residual limb length | prosthetic knee | prosthetic foot | K-Level | amputation dates back [y] |
|---|---|---|---|---|---|---|---|---|---|
| 1 | tumor | 172 | 62 | 33 | long | C-Leg [1] | C-Walk [1] | 3 | 10 |
| 2 | tumor | 186 | 81 | 47 | medium | C-Leg [1] | C-Walk [1] | 3 | 29 |
| 3 | trauma | 171 | 61 | 25 | medium | C-Leg [1] | C-Walk [1] | 4 | 7 |
| 4 | trauma | 175 | 85 | 52 | medium | C-Leg [1] | Axtion [1] | 3 | 26 |
| 5 | trauma | 183 | 100 | 23 | long | C-Leg [1] | C-Walk [1] | 3 | 8 |
| 6 | trauma | 176 | 87 | 51 | medium | C-Leg [1] | C-Walk [1] | 4 | 34 |
| 7 | trauma | 188 | 93 | 18 | medium | C-Leg [1] | C-Walk [1] | 3 | 1 |
| 8 | trauma | 190 | 93 | 34 | short | 3R60 [1] | Dynamic Motion [1] | 2 | 18 |
| 9 | bone abscess | 187 | 96 | 42 | long | C-Leg [1] | Variflex LP [2] | 3 | 2 |
| 10 | congenital deformity | 173 | 87 | 37 | long | C-Leg [1] | C-Walk [1] | 4 | 31 |
| 11 | trauma | 177 | 76 | 53 | long | Rheo Knee [2] | Ceterus [2] | 3 | 3 |
| 12 | trauma | 181 | 87 | 27 | medium | C-Leg [1] | C-Walk [1] | 3 | 2 |
| 13 | trauma | 178 | 70 | 53 | medium | 3A2000 [3] | Soleus [4] | 4 | 27 |

residual limb length in relation to the thigh length of sound side (<1/3 short, ≥1/3<2/3 medium, ≥ 2/3 long),

* body mass includes the mass of the prosthesis; Participant No.8 was excluded for further analysis, as his activity level, residual limb length and the use of walking aids differed considerably from the rest of the participants with TFA.

[1] = Otto Bock, Duderstadt, Germany;

[2] = Össur, Reykjavik, Iceland;

[3] = Streifeneder, Emmering, Germany;

[4] = College Park Inc., Warren, USA.

Prior to assessment, participants underwent a clinical examination to check their eligibility for the study. Inclusion criteria were: at least one year since amputation, able to perform a single leg stance on the sound side for at least five seconds without any assistance, sufficient residual limb length to attach the transducer strap during OpTIMo measurements, and no residual limb problems, e.g., pressure sores, edema, or pain within the last three months. Participants with TFA were classified from K-Level 2 to 4, according to the Medicare level functional classification system (MFCL) [30]. Only one participant (No. 8) was classified as K-Level 2, had a short residual limb and typically used crutches while walking outdoors (Table 1). During CGA he walked without the use of crutches. He further reported using the prosthesis less than three hours per day, while all others reported to use it more than three hours per day. He furthermore showed a strength deficit according to a modified Medical Research Council (UK) scale of 3+ (active movement against gravity with minor resistance) for hip abduction and flexion [31]. All other subjects showed normal strength with 5 out of 5 on this scale. Participant No. 8 was excluded from further analysis, due to these differences in prosthetic usage, residual limb length and the use of walking aids, when compared to the rest of the cohort. All participants reported that they were satisfied with their current prosthesis. Their habitual sockets were all total contact, ischial containment sockets without the use of liners. Participants had all obtained their sockets at the in-house prosthetics and orthotics department. Alignment of the socket was in accordance to Long's line in the coronal plane and considered possible hip flexion contractures in the sagittal plane [32, 33]. According to the documentation, none of the participants showed a hip-flexion contracture of the involved side. Prosthetic length was adjusted to achieve a levelled pelvis. This was controlled by equal height of the posterior superior iliac spines during quiet standing, with both limbs equally loaded. Prosthetic components were aligned according to the manufacturer recommendations and checked via L.A.S.A.R Posture device (Laser Assisted Static Alignment Reference, Otto Bock, Duderstadt, Germany), which projects the vertical component of the ground reaction force as reference either laterally or anteriorly onto the subjects lower limbs [34].

A group of 18 unimpaired subjects served as a reference group (REF; nine female; 44y±13y; 174cm±9cm; 71kg±12kg). The REF group underwent both OpTIMo testing and CGA. MIM and CGA moments were normalized to body mass.

## Gait analysis

For CGA, skin mounted markers were placed on all subjects (with TFA and REF) on anatomical landmarks according to the Plugin-Gait (Vicon, Oxford, UK) lower body marker set. Additional markers on the subjects' shoulder girdle (spinous process of the 7th cervical vertebrae, left and right acromion, and incisura jugularis) were used to observe trunk motion in relation to the global reference frame [35]. Participants were asked to walk on a 10m-long walkway at self-selected speed. A 12-camera system (Vicon, Oxford, UK) operating at 120 Hz and two force plates operating at 1080Hz (Kistler Instruments, Winterthur, Switzerland) were used to record data. Joint kinematics and joint kinetics were calculated using the conventional inverse dynamics approach with the Plugin-Gait software (PiG, Vicon, Oxford, UK) [36–39]. Joint internal moments were averaged across 5–10 trials with valid force plate hits. Prosthetic feet ankle markers were placed onto the foot cosmesis where the lateral malleolus is mimicked. The knee marker was placed on the axis of rotation for single axis joints and the upper, anterior axis for the two polycentric joints (Otto Bock 3R60 and Streifeneder 3A2000; Table 1). A thigh marker was placed laterally onto the socket. As kinetics were investigated during stance-phase while walking, it was decided to use standard CGA routines which do not allow for corrections of centres of gravity position, moments of inertia, or mass distribution of the

prostheses. As shown by Pamies-Vila et al., the changes of the anthropometric body segment parameters have little effects on gait analysis results during stance [40].

## Isometric muscle strength measurements

The OpTIMo device, used to determine isometric hip moments, consists of a rigid frame with a force transducer (Biovision, Wehrheim, Germany). During testing, the subject stood and was supported at the pelvis inside this frame (Fig 1A). Reflective markers on the pelvis and the sound limb were left in place during measurements.

The force transducer was attached via a strap and a lightly padded cuff to the thigh of the REF or to the residual limb, respectively. Individuals with TFA did not wear a socket during OpTIMo measurement. The cuff was placed at a similar height between groups, which was the upper third of the thigh or residual limb, respectively. The position of the force transducer and cuff was adjusted for each hip MIM direction (flexion, extension, abduction, and adduction / Fig 1B). During measurements, a vertical position of the thigh was obtained. Prior to measuring each motion direction, subjects had the opportunity to perform submaximal test trials with to become familiar with the setup and protocol. The number of trials for familiarization was not standardized. After familiarization with the test, participants performed two repetitions for each motion direction. The trials with the highest MIM for each direction were used for further analysis. Participants were verbally encouraged during the trial to pull on the force transducer strap with their maximum voluntary contraction for about 10 seconds. Transducer force data (16 bit resolution at 1080Hz) and marker trajectories (120 Hz) were captured simultaneously. To avoid fatigue becoming a confounding factor, participants were allowed to rest between measurements. Participants were instructed to pull the transducer strap only in the direction the hip was being tested (flexion-extension and adduction-abduction) to avoid combined movements, e.g., hip abduction with adjunctive hip flexion. Participants were asked to

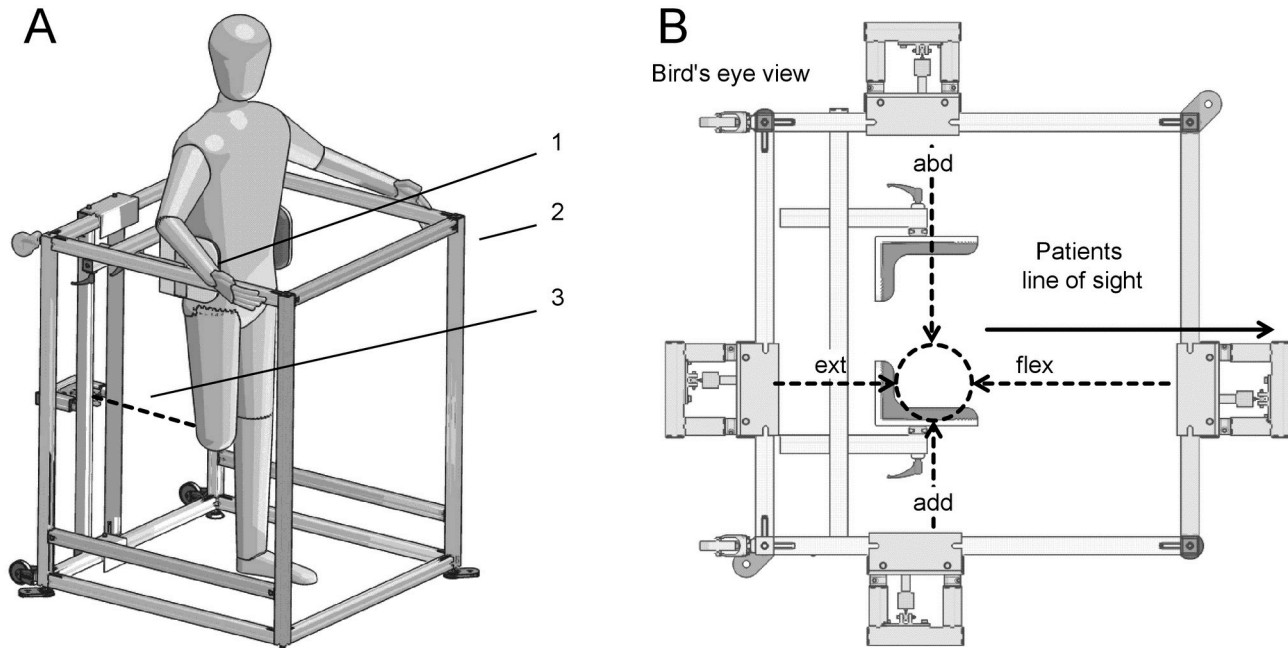

**Fig 1. OpTIMo (Optical Testing Isometric Moment) device to determine isometric hip moments in the study participants.** (A) Schematic representation, (B) Bird's eye view of the OpTIMo device with the different pulling directions during isometric tests. (1 = Pelvic Support; 2 = Frame; 3 = Force Transducer; / Modified Picture of [11] with permission).

constantly increase force during the trial and to attempt to hold their maximum for at least two seconds. The average force within this 2-second plateau was used for further data processing. The hip joint centres were determined by Davis regression equations [36] on the basis of the pelvic marker trajectories, similar to CGA routines (i.e., Plugin Gait, Vicon, Oxford, UK). The direction of force was determined by two markers and their associated trajectories, on the force transducer strap. The lever arm was defined by a vector from the calculated hip joint centre to a marker on the cuff. The isometric hip joint moment was subsequently calculated as the cross product of lever arm and force vector. A detailed description of the OpTIMo method can be found in Heitzmann et al. [11].

### Statistics

A Shapiro-Wilk Test was used to test for normal distribution of data. As some of the tested outcome variables were non-normally distributed, we chose non-parametric statistics (Mann-Whitney U Test for independent samples) to compare between groups (REF vs. TFA) and correlations between parameters within the participants with TFA (Spearman's rank correlation). To account for corrections for multiple comparisons (14 in CGA parameters and 4 in MIM) we applied a Bonferroni correction and adapted the levels of significance for MIM comparisons to $p < .0125$ and for CGA parameter comparisons to $p < .0036$.

Correlations with a Spearman's ρ of 0.40–0.59 were classified as moderate, 0.60–0.79 as strong, and 0.80–1.0 as very strong correlations. The level of significance for correlations was corrected to $p < .0036$ to account for testing across the coronal and sagittal plane CGA and MIM parameters.

## Results

We detected differences between the REF group and the group with TFA. In addition to the difference between groups in sex, we saw a significantly higher mean mass for the participants with TFA ($p = .025$). There were no statistical significant differences in height ($p = .087$) and age ($p = .346$) between groups.

The participants with TFA had a significantly lower MIM for hip abduction and adduction and hip flexion than the REF. They reached only 60% of the mean REF value in MIM abduction ($p < .001$), 63% in MIM adduction ($p = .001$) and 57% in MIM flexion ($p = .010$). There was no significant difference in hip extension MIM, with participants with TFA reaching 85% of the MIM of REF ($p = .215$). For temporal spatial parameters, individuals with TFA showed a comparable step length to REF. They walked slightly, but not significantly slower and had a significant decrease in cadence ($p < .001$) and step width ($p < .001$) (see Table 2).

For CGA kinematic results, participants with TFA showed a significant lower hip abduction and adduction range of motion on the involved side during walking. Furthermore, they showed significantly greater ranges for pelvic tilt, trunk obliquity and trunk tilt (see Table 1 / kinematics hip pelvis & trunk ranges 0–100% gait cycle). Additionally, they leant their trunk towards the involved side during single limb support, which was associated with an involved side pelvic drop and increased involved side hip abduction in stance (see Fig 2 and S1 Fig).

When compared to REF, participants with TFA had significantly lower peak internal hip abduction and adduction moments and a lower peak hip extension moment during the stance phase.

We detected significant strong, to very strong correlations between the hip MIMs in the different movement directions (abduction, adduction, extension, and flexion) in both groups (S2 Table). For hip abduction and adduction, MIMs in individuals with TFA showed a strong but not significant correlation. We saw no significant correlations between MIM and CGA

**Table 2. Results for the maximum isometric moments (MIM) and conventional clinical gait analysis (CGA).**

| | | Participants | | | | | | | | TFA vs. REF |
| --- | --- | --- | --- | --- | --- | --- | --- | --- | --- | --- |
| | | with trans femoral amputation (TFA) | | | | without impairment (REF) | | | | |
| | | (N = 12 mean of involved limb) | | | | (N = 18 mean of both limbs) | | | | |
| | | | | 95% CI | | | | 95% CI | | |
| | | mean | ±SD | LB | UB | mean | ±SD | LB | UB | |
| Hip MIM (N m/kg) | abd. | .85 | (±0.25) | .69 | 1.01 | 1.41 | (±0.41) | 1.21 | 1.62 | **p < .001*** |
| | add. | .87 | (±0.36) | .64 | 1.10 | 1.37 | (±0.42) | 1.16 | 1.58 | **p = .001*** |
| | ext. | 1.11 | (±0.33) | .90 | 1.32 | 1.30 | (±0.39) | 1.11 | 1.50 | p = .215 |
| | flex. | .93 | (±0.40) | .68 | 1.18 | 1.63 | (±0.73) | 1.27 | 1.99 | **p = .010*** |
| CGA Temporal spatial parameter | step length (m) | .73 | (±0.08) | .68 | .79 | 0.75 | (±0.07) | .72 | .79 | p = .602 |
| | Speed(m/s) | 1.22 | (±0.22) | 1.08 | 1.36 | 1.44 | (±0.17) | 1.35 | 1.52 | P = .008 |
| | cadence (steps/min) | 99 | (±9.94) | 93 | 105 | 114 | (±8.79) | 110 | 119 | **p < .001*** |
| | step width (cm) | 14.4 | (±3.63) | 12.1 | 16.7 | 6.9 | (±2.85) | 5.4 | 8.3 | **p < .001*** |
| CGA kinematics hip pelvis & trunk ranges 0–100% gait cycle (in degrees) | hip abd. & add. | 8.5 | (±3.25) | 6.5 | 10.6 | 13.7 | (±3.09) | 12.2 | 15.3 | **p < .001*** |
| | hip flex. & ext. | 44.1 | (±6.20) | 40.2 | 48.0 | 47.4 | (±4.36) | 45.2 | 49.6 | p = .134 |
| | pelvic obliquity | 7.9 | (±2.82) | 6.1 | 9.7 | 9.5 | (±2.23) | 8.4 | 10.6 | p = .104 |
| | pelvic tilt | 8.2 | (±1.69) | 7.1 | 9.3 | 2.4 | (±1.27) | 1.8 | 3.1 | **p < .001*** |
| | trunk obliquity | 5.9 | (±2.11) | 4.6 | 7.3 | 2.7 | (±0.83) | 2.3 | 3.2 | **p < .001*** |
| | trunk tilt | 3.9 | (±0.98) | 3.2 | 4.5 | 2.0 | (±0.76) | 1.6 | 2.4 | **p < .001*** |
| CGA maximum hipmoments in stance (N m/kg) | abd. | .59 | (±0.19) | .47 | .71 | 0.88 | (±0.18) | .79 | .96 | **p = .001*** |
| | add. | .06 | (±0.04) | .04 | .08 | 0.16 | (±0.10) | .10 | .21 | p = .009 |
| | ext. | .46 | (±0.16) | .35 | .56 | 1.07 | (±0.26) | .94 | 1.19 | **p < .001*** |
| | flex. | 1.06 | (±0.29) | .88 | 1.25 | 0.88 | (±0.21) | .77 | .98 | p = .059 |

TFA = transfemoral amputation; REF = group of unimpaired peers which served as a reference; CI = 95% confidence interval of the mean; LB = lower bound;
UB = upper bound; SD = standard deviation; level of significance for CGA parameters p < 0.0036; for MIM p < 0.0125

parameters for the coronal or sagittal plane, in those with TFA. However, while not significant, we saw a moderately negative correlation between hip abduction MIM and trunk obliquity range (ρ = -0.45; p = 0.14) in participants with TFA. Lastly, a moderately negative correlation was found for TFA hip adduction MIM and hip adduction moments during walking (ρ = -0.63; p = 0.03), but also a moderately positive correlation between hip flexion MIM and the corresponding moments during gait (ρ = 0.49; p = 0.11). There was no correlation between peak hip abduction moments in stance and the corresponding abduction MIM (ρ = -0.032; p = 0.92), nor did we find a correlation between peak hip extension moment in stance and the extension MIM (ρ = 0.042; p = 0.90) for participants with TFA.

## Discussion

Our data shows that those individuals with TFA have significantly lower MIM values for hip abduction, adduction, and flexion than the REF group, which supports our first hypothesis that people with TFA show significant strength deficits on the involved side. These findings are generally in line with the literature, e.g. hip joint muscles strength deficit in people with TFA of up to 35% when compared to unimpaired peers [7, 10, 41].

In addition, we detected several gait deviations in the cohort with TFA. A significantly higher mean range in trunk obliquity (mean 5.9±2.11˚) over the entire gait cycle was seen in

**Fig 2. Schematic coronal plane views presented during left and right single support (representative trial of a participant with TFA and a REF participant).** The participant with TFA presents a lean of the trunk to the involved side with a linked pelvic drop and increased hip abduction on the involved side.

comparison to REF (2.7±0.83°) (Table 2, S1 Fig). This was accompanied by an increased hip abduction, indicated by the decreased range in abduction and adduction (Table 2, S1 Fig). The participants with TFA appear to avoid hip adduction. Instead, they possibly shifted their mass by means of the increased lateral trunk obliquity to the involved side. This compensation may explain the decreased hip internal abduction moments on the involved side in participants with TFA (S1 Fig), as has been observed in people with TFA [25, 26, 42, 43]. Occasionally this pattern in people with TFA is referred to as being a Trendelenburg gait pattern. However, we do believe that this not correct, as Trendelenburg's original publication described a different pattern, i.e. a pelvic drop on the swinging limb, in a different pathology [21]. In contrast to Trendelenburg's explanations, we monitored a coronal plane pelvic drop in the affected side single-support-phase with a linked pelvic lift on the contralateral side, and an accompanied trunk lean to the affected side (Fig 2). Tazawa et al. indicated that subjects with a subjectively 'good' walking pattern showed a smaller lateral trunk displacement [43]. Altered trunk and hip kinematics exhibited in persons with TFA have been attributed to weak hip musculature [14, 25, 42]. This connection may explain the observed, non-significant, moderate negative correlation between hip abduction MIM and trunk obliquity range-of-motion. This correlation indicates that weaker people with TFA have the tendency to show a greater trunk range-of-motion

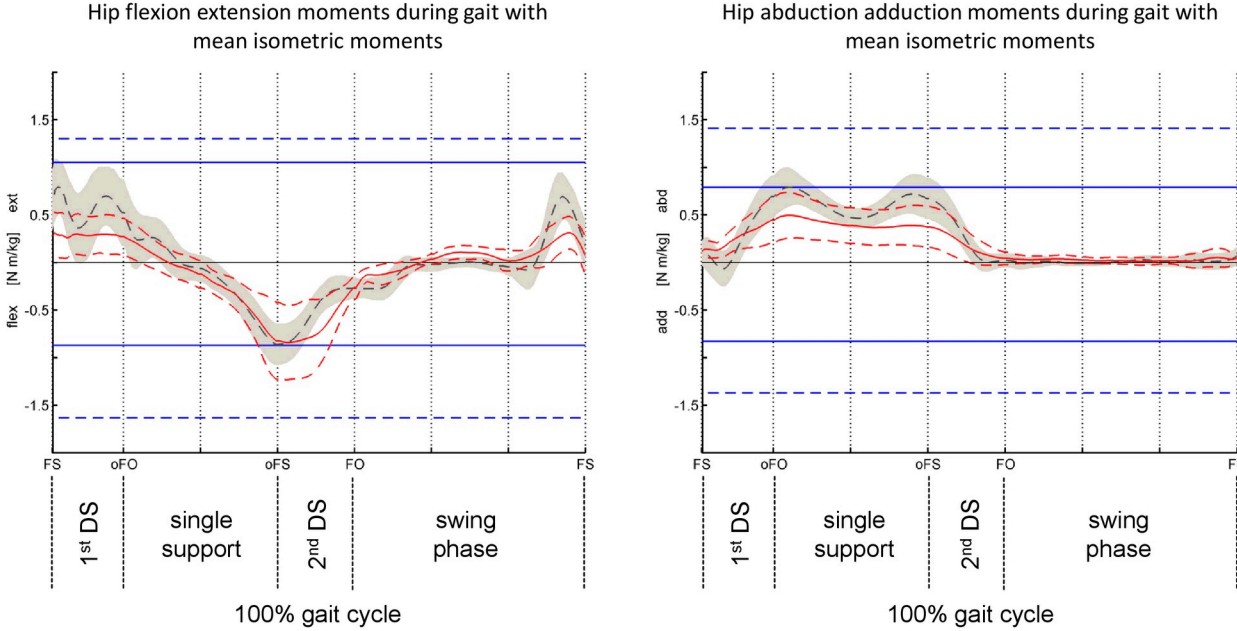

**Fig 3. Sagittal and coronal plane hip moments during walking (plotted against the gait cycle) in comparison to the corresponding isometric moments collected during isometric muscle tests.** The scalar MIM values are plotted as horizontal lines for comparison to CGA moments, similar to Fosang et al. [27]). red, solid line = mean moments during walking of participants with TFA, walking @ 1.22±0.22 m/s; red, dashed line = standard deviation of the TFA mean; grey, dashed line = mean moments during walking of REF @ 1.44±0.17 m/s; grey band = standard deviation of the REF mean; blue, solid line = mean, involved side MIM of participants with TFA; blue dashed line = mean MIM of REF; Legend for events and subphases of the gait cycle: FS = foot strike; oFO = opposite foot of; oFS = opposite foot strike; FO = foot off; DS = double support.

during walking than their stronger piers. However, it should be noted that this is the exact opposite trend observed in the unimpaired population (S2 Fig), showing a moderate positive correlation. The only significant and strong, to very strong correlations were found between the MIMs of different movement directions with each other (S2 Table and Scatterplots S3 and S4 Figs). Thus, a weaker person shows generally lower hip MIM values, which was true in both TFA and REF groups.

Peak to peak ranges in MIM were significantly smaller in participants with TFA (Fig 3). In theory, one could imagine these peak to peak values as a working range. If this working range would be surpassed by moments occurring during gait, this may lead to gait deviations. In typically developing children, the hip MIMs were about 2 to 3 times as high as the moments during walking [27]. At the same time their ankle moments during walking exceed their ankle MIM [27]. Therefore, the model of the peak to peak MIM values building a "working range" certainly has limitations and does not fully explain the relationship between monitored strength deficits and gait deviations in people with TFA. In REF and individuals with TFA, we observed MIM values 1.5 to 2 times higher than the peak moments during gait. This was also true for participants with TFA as they had generally reduced moments during walking, which led to a similar ratio. Although, the aforementioned "working range" model and the relationship between isometric moments and moments during walking are not fully understood, those with TFA may pursue a similar ratio of MIM and peak moments during walking as REF, potentially to reach comparable "headroom" as their unimpaired peers. The price for this strategy may be gait deviations which lead to reduced moments during walking. However, this is a conjecture and we cannot fully support this theory with the data collected in this study. Likewise, it was concluded that more work is required to understand how the MIM measurements

relate to moments during walking [10]. Additionally, participants with TFA walked considerably, but not significantly slower, which makes comparisons of kinetics difficult.

In addition to the strength status of the involved side, other confounding factors will influence gait quality in individuals with TFA. Such factors could be technical characteristics of their prosthesis such as socket design, prosthetic alignment, potential leg length discrepancies and prosthetic components. The lack of any significant correlation that corresponded directly to a gait deviation supports our suggestion that gait deviations in our TFA cohort are multifactorial. Especially, in our opinion, the socket plays a crucial role. The interactions between the residual limb and the socket will result in a pseudo-joint, independent of the chosen socket design. A reduced power transmission between the stump and the socket may cause the observed muscle weakness to manifest a more pronounced functional deficit. We believe there is an accumulation of these two factors, i.e. the structural changes of the limb and the reduced power transmission. Some socket designs may be even more prone to increased residual limb-socket interactions, when compared to others. For example, it is well accepted that quadrilateral sockets do not stabilize the femur in the coronal plane and therefore will more likely cause increased residual limb movement during walking, compared to other designs, e.g. ischial containment sockets [44, 45]. Decreased medio-lateral stability of the residuum, along with other cofounding factors like strength deficits of the hip abductors and a leg length discrepancy, was identified as a potential cause for a lateral trunk motion [46]. The strength status of people with TFA is therefore of high clinical relevance. Strength deficits may force people with TFA into compensation mechanisms during walking, which should be distinguished from those induced by the socket, the prosthetic components, or leg length induced gait alterations. Moreover, the affected side ground contact in a person with TFA is different than in an unimpaired person. In people with TFA the affected side hip joint is the only joint which can be volitionally controlled and therefore used to adjust the loads introduced to the prosthetic limb. Thus, gait deviations in this population may be a response to unwanted loads introduced to the involved side, and may serve as an option to readjust loads. Another indicator, that coronal plane gait deviations cannot purely be attributed to hip strength deficits, is that a lateral trunk lean can be also present in amputations below the knee. So, there is a high plausibility that not all gait deviations, in particular of the pelvis and trunk can entirely be lead back to weaknesses of the hip.

Hence, the origins for strength deficits in people with TFA and their gait deviations, partially attributed to these strength deficits, cannot be fully clarified in this study. For example, the hip abductors are not directly damaged by the transfemoral amputation per se. However, the muscle balance between abductors and adductors is disturbed. The adductors' moment arm is reduced, which leads to increased residual limb abduction, which will affect the abductor moment arm. As a result, the altered muscle loads may be insufficient to maintain strength of the hip abductors. As stated earlier, the altered biomechanics of prosthetic gait in subjects with TFA may induce a weakening of certain muscle groups, while possibly also preserving strength of other muscle groups. This hypothesis is partly supported by the hip extension MIM results in our TFA group. It is noteworthy that we detected no significant strength deficit in hip extension MIM in individuals with TFA, which, to our knowledge, has not been reported previously. This was the only MIM parameter where we did not find a significant weakness, which could be explained by the fact that the hip extensors are important to provide propulsion and an upright posture in those with TFA but also to stabilize the prosthetic knee joint in early stance, mainly in conventional, non-microprocessor controlled prosthetic knee joints. The subjects seem to utilize an increased internal hip extension moment, which may be sufficient to preserve strength in the hip extensors of our, higher active cohort with TFA. This is also in line that hip extensors, of both the involved, and the sound side, compensate for the loss of function of the involved side [47]. Reduced hip extension power on the involved side

has further a negative impact on preferred walking speed [48]. So, individuals with TFA seem to utilize their hip extensors more than other muscle groups of the residual limb.

In the context of this study, we must also report its main limitations. Prosthetic alignment was controlled in the in-house workshop but was not quantified furthermore. Although the prosthetic leg length was checked, in that the pelvis was levelled in quite standing, the prosthesis may be shorter in single limb support, e.g. due to pistoning or due to compression of elastic prosthetic feet. As mentioned previously, deficits in alignment and leg length discrepancies lead to gait deviations, also of the trunk. Socket comfort was not determined in this study [49]. A socket that does not fit comfortably could aggravate gait deviations, e.g. the occurrence of an increased pistoning or a reduced loading to avoid pain. However, all participants reported being satisfied with their current socket and prosthesis in general. Generally any convenient sample, of either unimpaired or TFA participants, is limiting. The sample size in this study was small and may have influenced the results, in particular significance levels of correlations between MIM and GCA parameters. However, we do believe that the moderate correlation between abduction MIM and lateral trunk motion could be an indication that strength is one of the cofounding factors in lateral trunk displacement of individuals with TFA during walking. The rather high activity level or K-Level of our TFA group may also influence results. Low activity individuals with TFA (i.e. K-level 1 and 2) may show even greater deficits in strength and larger deviations in gait. The differences in age and gender between REF and TFA groups may also represent a limitation. In this context, it is remarkable that the potentially "stronger" participants with TFA (younger and only male participants) are still significantly weaker in three out of four of the hips MIM in comparison to REF. People with TFA walked slower. This difference was not significant. However, it is noticeable and may bias the interpretation of kinetics.

## Conclusion

The relation between maximum moments during gait and the corresponding MIM might be helpful in detecting strength related compensation mechanisms. However, during walking compensation mechanisms may also be induced, e.g. by the socket, prosthetic leg length or prosthetic component related, that contribute to the net functional deficit. Though, we have demonstrated that there are moderate correlations between gait deviations, i.e. later trunk lean onto the involved side and hip abductor MIM. However, this was not significant and was the only correlation that corresponded directly to a gait deviation. This supports our suggestion that gait deviations in our cohort of individuals with TFA are multifactorial. Future studies should investigate additional influential factors e.g. by comparing different socket designs and further explore the relationship between MIM and kinematics and kinetics during gait.

## Supporting information

**S1 Table. Self-reported health status and sports activities of the participants with TFA.** # = nothing specified.
(DOCX)

**S2 Table. Correlations of sagittal and coronal plane gait parameters with their corresponding maximum isometric moments (MIM) of the hip.** Values Spearman's rho with its corresponding p-value; level of significance was set to $p < .0036$; * = statistical significant correlation; # = moderate, + = strong, ! = very strong; correlations with a Spearman's $\rho$ of

0.40–0.59 were classified as moderate, 0.60–0.79 as strong, and 0.80–1.0 as very strong correlations).
(DOCX)

**S1 Fig. Trunk, pelvic and hip coronal and sagittal kinematics as well as coronal and sagittal hip kinetics (internal moments) of both investigated cohorts.** Red, solid line = mean kinematics and kinetics during walking of participants with TFA walking @ 1.22 (±0.22) m/s; red, dashed line = standard deviation of the TFA mean; grey, dashed line = mean moments during walking of REF @ 1.44(±0.17) m/s; grey band = standard deviation of the REF mean; FS = foot strike; oFO opposite foot of; oFS = opposite foot strike; FO = foot off; DS = double support.
(TIF)

**S2 Fig. Scatterplot of the isometric trunk obliquity range of motion vs. hip abduction moment during the gait cycle.**
(TIF)

**S3 Fig. Scatterplot of isometric coronal plane hip moment against each other.**
(TIF)

**S4 Fig. Scatterplot of isometric sagittal plane hip moment against each other.**
(TIF)

## Acknowledgments

The authors would like to thank all study participants. Their contribution is highly appreciated. We further like to thank Dr. Alan R. De Asha for editing the manuscript.

## Author Contributions

**Conceptualization:** Sebastian Immanuel Wolf, Merkur Alimusaj.

**Data curation:** Daniel Walter Werner Heitzmann, Michael Günther, Sebastian Immanuel Wolf, Merkur Alimusaj.

**Formal analysis:** Daniel Walter Werner Heitzmann, Julien Leboucher, Michael Günther, Cornelia Putz, Marco Götze, Sebastian Immanuel Wolf.

**Funding acquisition:** Sebastian Immanuel Wolf, Merkur Alimusaj.

**Investigation:** Daniel Walter Werner Heitzmann, Julia Block, Michael Günther.

**Methodology:** Daniel Walter Werner Heitzmann, Julia Block, Sebastian Immanuel Wolf.

**Project administration:** Daniel Walter Werner Heitzmann, Julia Block, Sebastian Immanuel Wolf, Merkur Alimusaj.

**Resources:** Sebastian Immanuel Wolf.

**Software:** Daniel Walter Werner Heitzmann.

**Validation:** Daniel Walter Werner Heitzmann.

**Writing – original draft:** Daniel Walter Werner Heitzmann.

**Writing – review & editing:** Daniel Walter Werner Heitzmann, Julien Leboucher, Julia Block, Michael Günther, Cornelia Putz, Marco Götze, Sebastian Immanuel Wolf, Merkur Alimusaj.

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
