## [Decision Letter · Decision Letter 0]

4 Dec 2019

PONE-D-19-24706

The influence of hip muscle strength on gait in individuals with a unilateral transfemoral amputation – a prospective cohort study

PLOS ONE

Dear Mr. Heitzmann,

Thank you for submitting your manuscript to PLOS ONE. After careful consideration, we feel that it has merit but does not fully meet PLOS ONE’s publication criteria as it currently stands. Therefore, we invite you to submit a revised version of the manuscript that addresses the points raised during the review process.

We would appreciate receiving your revised manuscript by Jan 18 2020 11:59PM. To enhance the reproducibility of your results, we recommend that if applicable you deposit your laboratory protocols in protocols.io, where a protocol can be assigned its own identifier (DOI) such that it can be cited independently in the future. For instructions see: http://journals.plos.org/plosone/s/submission-guidelines#loc-laboratory-protocols

We look forward to receiving your revised manuscript.

Kind regards,

Yih-Kuen Jan, PhD

Academic Editor

PLOS ONE

Journal Requirements:

'The authors have declared that no competing interests exist.'

We note that one or more of the authors are employed by a commercial company: 2Guenther Bionics GmbH.

4. Please include a copy of Table 3 which you refer to in your text on page 20.

Additional Editor Comments (if provided):

Reviewers' comments:

Reviewer's Responses to Questions

**Comments to the Author**

1. Is the manuscript technically sound, and do the data support the conclusions?

Reviewer #1: Yes

Reviewer #2: Yes

Reviewer #3: Yes

2. Has the statistical analysis been performed appropriately and rigorously? 

Reviewer #1: No

Reviewer #2: Yes

Reviewer #3: Yes

3. Have the authors made all data underlying the findings in their manuscript fully available?

Reviewer #1: Yes

Reviewer #2: Yes

Reviewer #3: Yes

4. Is the manuscript presented in an intelligible fashion and written in standard English?

Reviewer #1: Yes

Reviewer #2: Yes

Reviewer #3: No

5. Review Comments to the Author

Reviewer #1: The manuscript reports results from a study examining the changes in gait kinematics and joint moments that occur in a cohort of participants with transfemoral amputations (pTFA), and their relationship to isometric joint moments. To do this, the authors use an isometric force measurement rig in conjunction with marker based clinical gait analysis to compare maximum isometric joint moments (MIM) with joint moments and kinematics measured during walking. The experiment design appears solid (although clarifications are needed), and the results show consistent reductions in maximum isometric moment for hip abduction, adduction, and flexion, correlations between isometric and gait joint moments, and a moderate correlation between isometric hip abduction moment and trunk movement. As the authors point out in the discussion, many of the findings agree with previous studies looking at isometric moment generation or gait kinetics/kinematics in pTFA, although here the authors are also able to examine relationships between the two metrics.

Overall the manuscript is well written. There are a variety of minor issues that require correction or clarification in the manuscript but most are relatively easily addressed. Clearly this is a rich dataset with a wide number of measures obtained during testing, particularly during gait. It looks like table 2 lists all of the measures examined but it would be helpful for the reader if they were stated more explicitly in the methods, possibly as a list of the planned study measures before they are detailed in the main text. The nonparametric statistical testing for group differences is appropriate, however, it is not clear whether corrections for multiple comparisons were applied to help control for false positive results given the number of tests.

The Discussion is quite long, which is not necessarily a problem, although there are a number of associations made between pTFA and other populations (e.g. hip luxation, cereberal palsy, Legg Calve’ Perthes, etc.), for which the rationale is not always clear given the inherent differences between such groups.

Page 6 (lines 116-121), The point the authors are trying to make with Sherk et al. is not very clear.

Page 12 (line 249), please clarify why adjustments for the characteristics of the prosthesis were not made when calculating joint kinetics. This would seem likely to have an impact on the measures.

Page 13 (line 266), references Figure 1C which is not present in Figure 1.

(line 271), notes that marker trajectories were measured during isometric testing but their use is not clarified later on.

(line 287) indicates that the “best trial” was used. Please clarify how this was defined.

Figure 3, The content of the figure is not clear and the caption does not explain all elements of what is being presented. For example, the titles suggest an x vs. y comparison of the title variables in the plots but the independent variable appears to be gait cycle (although abbreviations are not defined). Shading and dotted lines are not defined in the caption. Please also doublecheck the phrasing ‘none continual scalar’.

Page 19, reports the spearman correlations between various metrics in TFA but it is unclear how the levels of correlation relate to controls, which are not presented.

Page 22 (lines 425-427), if the participants maximum isometric joint moment was being measured in the study (which generally speaking should be the largest moment that can be generated), it is unclear how/why moments generated during gait would necessarily exceed it.

Figure S1 caption, please indicate what the ranges correspond to.

Other minor comments and suggested edits (insertions/deletions in brackets),

Please verify that past tense grammar is consistent throughout the manuscript

Page 4 (line 84), “… might [be] helpful…”

Page 6 (line 127), “…Additional[ly]…”

(line 132), “…compromise[d]…”

(line 135), “…who [was] utilizing…”

Page 7 (line 156), “…walking [is] coronal plane…”

(line 159), “…and [] irregularities…”

(line 161), “…TFA [out] of a ….”

(line 164), “…this may [also] be related to…”

Page 8 (line 179), “…role of [muscle] strength…”

Page 9 (line 190), delete “of the department”

Page 16 (line 325-326), “…vie [of] each side…present[ed]…”, “conjunct” spelling?

Page 21 (line 390), “…patter[n]…”

Page 22 (line 426), “…if this [] working…”

Page 23 (line 445), “…socket do[es] not…”

Reviewer #2: The goal of the manuscript is to examine the influence of hip muscle strength on walking biomechanics for persons with transfemoral amputation. The authors hypothesize that: “people with [transfemoral amputation] TFA will show lower isometric moments on the involved side than those in reference a group (REF); we expect correlations between maximum isometric joint moments and gait kinematics of the hip, pelvis, and trunk; and, further, we expect correlations between maximum hip joint moments in gait with those tested isometrically.” The authors prove their hypothesis and show that persons with transfemoral amputation have reduced hip muscle strength as compared to healthy individuals, but do not attempt to decouple the effects of reduced hip muscle strength and differences in leg length due to the transfemoral prosthesis for the gait deviations observed. It is recommended that the authors perform additional analyses to examine the role of the transfemoral prosthesis on the gait deviations observed rather than ignoring this confounding factor.

Major:

In the discussion and conclusion, the authors link decreased hip ab/adduction moment to changes in hip and trunk kinematics. While this may be a factor, the reviewer does not believe it is the cause of the kinematic changes. Most prosthetists make create a transfemoral prosthetic system that causes the prosthetic limb to be longer than the sounds limb so that during walking the user compresses the prosthetic foot during stance. This can lead poor hip alignment during standing and during the stance phase of walking if the user does not meaningfully progress onto the prosthesis during this time. Additionally, this longer prosthetic leg can lead to hip hiking during swing. Did the authors examine the trunk obliquity and hip alignment during a static trial? Additionally, the authors could examine the effective leg-length throughout walking by examining a vector from the hip joint center to the center-of-pressure under the foot. Such a vector should be compared to the sound limb to see if the effective limb length is the same. This would allow the authors to make a more concrete statement about how the subjects interact with the prosthesis and if the observed hip and trunk kinematic changes are due to leg length changes or due to weak hip musculature.

The introduction and discussion are clunky. The authors often introduce a source and then explain what the study concluded (or the inverse). For example:

“Pröbsting et al. gathered studies on changes in the musculoskeletal system as a result of lower-limb amputation [1]. They categorized these musculoskeletal changes as follows: low back pain, osteoarthritis, reduced bone density, and muscle atrophy.”

This type of referencing occurs 15+ times throughout the introduction and reduces the readability of the article. It is encouraged for the authors to concisely revise the introduction and discussion. An example for the previous quote would be:

“Noted musculoskeletal comorbidities following lower-limb amputation are: low back pain, osteoarthritis, reduced bone density, and muscle atrophy [1].”

Make sure to check the grammar and punctuation throughout the manuscript. The reviewer has listed several grammar and punctuation items that need to be addressed in the introduction but did not do so for the entire manuscript.

Minor:

This is not a prospective study as there was not an extended period of time that the subjects were examined over. Please remove such wording from the title and the manuscript.

Make sure to have continuous past tense throughout the manuscript

Introduction:

Line 91: need a comma instead of and between “integrity” and “perception”

Line 102: “this” to “muscles strength” as in the previous sentence you refer to muscle atrophy, amputation and strength

Line 109: “Medial” instead of ventral? The reviewer does not know what the ventral side of the thigh is.

Line 114: add “muscles” to the end of the sentence.

Line 127: “Addition” to “Additionally,”

Line 132: “compromise” to “compromised”

Line 138: “Additionally”

Line 156-157: “A well-known example of how strength deficits impact walking are probably coronal plane gait deviations of the trunk, as described by Trendelenburg [23].” Is not a definitive sentence.

Line 178-179: Remove: “We are confident that by using this approach we will gain better insight into the role of strength in people with TFA and their compensation mechanisms in gait.”

Methods:

How much does the K2 subject skew the results? Should he/she be eliminated from subsequent analyses? This subject, as reported, does not spend as much time on the prosthesis as the other subjects and is the only subject collected that was classified as K2.

Lines 190-191: remove “of the department”

Line 217: Define the acronym MRC

Lines 221-222: Were these sockets different than their daily use sockets? If they were, how much do you think this affects the gait outcomes of the manuscript?

Line 225: Define the acronym LASAR

Lines 249-250: How does not adjusting for the prosthetic inertial parameters affect the joint kinetics? And how does this affect the comparisons the authors make between able-bodied and persons with transfemoral amputation?

Line 266: a comma is needed

Line 288-291: The reviewer does not understand the how this portion is relevant. What was being correlated to provide a high correlation?

Line 289: put (ICC) after interclass correlation coefficient

Line 294: Please define PTFA acronym.

Results:

Why compare peak-to-peak maximum isometric moments? This would seem to penalize persons with amputation or hip musculature weakness more. Additionally, this does not provide an accurate characterization of the musculature which is only active in one direction. It is strongly recommended to separate out different motions as is reported in Lines 352 to 355 of the discussion and to not use the peak-to-peak range.

Line 313: include “during walking” to the end of the topic sentence.

Table 2: It is encouraged to not statistically compare gait parameters between the persons with amputation and healthy individuals as they had a significantly different gait velocity. It has been shown that differences in gait velocity cause differences in lower limb kinematics and kinetics (see Zelik and Kuo 2010).

Figure 3: Please update the healthy line color as the reviewer cannot see the mean line.

Figure 3 caption: please include the mean walking velocities for the persons with amputation and healthy individuals.

Line 322: this was not a significant finding. why report without significance?

Can the authors include a table with the variables correlated, the correlation values and the significance of the correlation values?

Discussion:

Line 350: “Individuals”

Lines 355-358: The reviewer does not see the relevance in reporting the ranges in the MIM. See comment above.

Line 363: “und” to “and”

Lines 368 to 438: Paragraph may want to be broken up to improve readability. Also, the authors refer to many different clinical populations (ex: children with Legg Calve´ Perthes disease, persons with cerebral palsy). This provides un-useful information as many clinical populations may suffer from different co-morbidities which makes the population unique and less relatable to each other.

Lines 411-420: Seems to be reiteration of the results with no real discussion going on. Additionally, are all the correlation values significant that are cited?

Line 425: “In theory one could imagine these peak to peak values as a working range and if this this working range will be surpassed by moments occurring during gait this may induce gait deviations or even make it impossible to walk.” These peak to peak ranges are exceeded during human running (see Riddick et al. 2018) for healthy individuals. The reviewer does not feel that this hypothesis can be supported as human locomotion exceeds these thresholds. It is recommended to temper the language.

The review thinks that the Author’s statement of: “Strength deficits may force individuals with TFA into compensation mechanisms, which should be distinguished from socket or prosthesis-induced gait alterations. External factors inducing the significant strength deficits in people with TFA cannot be fully identified in this study.” is the crux of the article and should be stated earlier in the discussion than line 462.

Line 466-467: “The hip abductors are not directly weakened by the transfemoral amputation itself.” is a misleading statement. The hip abduction muscles may not loose muscle mass during the amputation BUT the thigh muscles are “tacked down” onto the femur or may not even be attached. In the former case, the muscle loses the moment arm at which it acts, effectively making it weaker to perform the necessary task.

Line 470-481: Why do the authors interpret non-significant results as significant results?

Conclusion:

The authors state that this information may be helpful for rehabilitation. The authors should discuss such implication in the discussion.

Reviewer #3: This manuscript investigates potential correlations of strength gait deviations, as well as compared a cohort of people with transfemoral amputation to people without amputation. The topic of this article is interesting because if we could see a gait deviation and then address it with rehabilitation (e.g., strength or balance), it would help people with amputation walk with less deviations. I have a series of comments below that reference major and minor concerns, and I have highlighted some of them with line numbers but did not highlight every instance that is repeated throughout the manuscript.

General questions: Is maximal strength important with over ground preferred walking speed, or is it more important to show how much of their overall capacity they need to use to walk compared with non-amputees? Is there a greater reserve strength for non-amputees? Maybe a deviation is chosen to save available muscle fibers and reserve capacity to respond to perturbations or minimize energy expenditure.

Major Comments

Abstract:

Line 51: Is this a prospective study? A prospective study identifies a population and then follows them and tries to make predictions about what will happen, whereas here you are trying to make correlations between gait deviations and strength deficits.

Lines 51-53: Aside from the prospective study comment, I like this statement because it is clear about what you did in this study.

Introduction:

Line 95: This is something done many times throughout the manuscript, and may just be a personal preference on my part, but other authors should be cited while their findings are prominent. “Probsting…” sentence adds nothing. Then line 104, “Ostchega et al….” did not prove a relationship, the results or findings of a study show something, the authors just write it up. Also, there are very few proven relationships, there is more evidence for some things than others, but one study (and multiple studies) does not prove a relationship.

Line 102: Unless you plan on talking about other forms of mobility, such as transfers, just say “this relates to walking.”

Throughout the introduction there could be more standalone paragraphs, and rather than listing findings from study after study, the information could be synthesized or summarized. This relates to my comment above about authors not showing anything, but the overall data collected from many studies.

Lines 173 and 174: There should not be methods in the introduction.

Lines 182-183: Should hypothesized directionality of correlations, not just state there will be correlations.

Methods:

Line 213-215: I am not sure if participant 8 should be included in the data analysis, especially since he is a K2 ambulator and then he walked without crutches during his motion analysis. Walking without crutches is not his normal or preferred, and therefore the gait speed and mechanics are not preferred, whereas the participants were using preferred mechanics.

Lines 267-268: Did you standardize the amount of test trials for participants to try the OpTIMo?

Lines 284: General questions about OpTIMo, how widespread is this device clinically or in research? Is it a research-only device? How does it compare to typical clinical force dynamometry measures with patients laying down?

Line 287: What makes a trial the best?

Line 295: Why not look at the sound side too? Could be interesting when comparing to affected side and unaffected participants.

Statistics look right.

Results:

Line 310-311: All the differences were significant except for the step length.

Figure 3: Should label x and y axes rather than leave that to the title. Also, I do not know what FS, FO, oFS, and oFO are, I assume they are foot strike and foot off, but do not know.

Lines 342-345: What does negative and positive correlations mean in your results?

Discussion:

General comment: Should not repeat results, especially with p-values, in the discussion section, they should be in the results section alone.

Line 375: Remove distinguished, that is your spin and unnecessary.

Line 382: The results from Cappozzo agree with your results, do not confirm your results, and if anything, your results confirm their finding.

Lines 409-421: Listing many results in the discussion section, could be synthesized with the literature better.

Minor Comments

Abstract:

Line 47: Awkward wording “on the person concerned”

Line 59: should be “Kinematics” not “Kinematic”

Line 59: Could say determined, not derived, unless you derived equations.

Line 72: Could say “were identified, i.e. significantly lower speed,…”

Introduction:

Line 91: Could say “body integrity, perception…”

Line 98: “clear atrophy” does not make sense, atrophy alone works

Line 98: I recommend saying amputated or affected side, rather than ipsilateral side. I understand ipsilateral but I am not sure every reader will, it is more clear to say something like amputated, affected or involved.

Line 102: “particularly” does not add anything, like most adverbs

Line 126: What do you mean by “redundant”?

Line 127: Could say “Additionally, the…”

Line 128: What do you mean by “finite coupling”?

Line 130: Could say “Gholizadeh et al. underline the problems in…”

Line 138: Could say “Additionally, new…”

Line 164: Lower case i in ischial

Methods:

Line 193: clinical gait analysis can be replaced with CGA because the acronym is already listed in the beginning

Line 225: Is the L.A.S.A.R. posture device widely known? I am not sure if it should be explained more or you could stop the sentence with “recommendations.”

Line 240: What was the collection rate for the force plates?

Results:

Line 315: “in the patients” is awkward wording

Lines 320-321: “For CGA…the patient group.” is not a clear sentence, could be written better.

Line 325: Why is this schematic “exemplary”?

In table 2, Kg should be kg, and it should be N m and not Nm because they are 2 separate words.

Line 339: Should say “with TFA hip MIM of…” and not “with TFA Hip MIMof…”

Discussion:

Line 390: pattern, not patter

Line 445: Say For example, no E.g. to lead a sentence

Line 445: does rather than do

Line 449-452: Not a clear sentence.

Line 452: stabilize rather than stabile

Line 501: address rather than tackle deficits, tackle is in American football or Rugby rather deficits of our patients/clients

6. PLOS authors have the option to publish the peer review history of their article (what does this mean?). If published, this will include your full peer review and any attached files.

Reviewer #1: No

Reviewer #2: No

Reviewer #3: No

---

## [Author Response · Author response to Decision Letter 0]

5 Jun 2020

General response from the authors: We would like to thank all reviewers for their time and effort when reviewing this manuscript. The input of the reviewers substantially helped us to improve the manuscript. We have tried to answer all raised questions adequately. Further we tried to shorten the manuscript and be more concise in describing effects. We excluded one critical patient from the cohort and added additional calculations (REF correlations and scatterplots of the correlated parameters). We do apologize for the almost unreadable tracked changes version, due to the many changes. We further involved a native speaker and fellow researcher in the field of prosthetics and orthotics for language editing of the manuscript. Also, since we did rewrite major parts of the manuscript, some of the specific critique of the reviewers may not apply any more, as we deleted or changed the sentence concerned. We apologize when this is sometime annoying. We hope for a positive feedback of the reviewers.

Reviewer #1: The manuscript reports results from a study examining the changes in gait kinematics and joint moments that occur in a cohort of participants with transfemoral amputations (pTFA), and their relationship to isometric joint moments. To do this, the authors use an isometric force measurement rig in conjunction with marker based clinical gait analysis to compare maximum isometric joint moments (MIM) with joint moments and kinematics measured during walking. The experiment design appears solid (although clarifications are needed), and the results show consistent reductions in maximum isometric moment for hip abduction, adduction, and flexion, correlations between isometric and gait joint moments, and a moderate correlation between isometric hip abduction moment and trunk movement. As the authors point out in the discussion, many of the findings agree with previous studies loing at isometric moment generation or gait kinetics/kinematics in pTFA, although here the authors are also able to examine relationships between the two metrics.

Response from the authors: We thank the reviewer for the thorough summary and hope that we can answer all raised questions sufficiently. 

Overall the manuscript is well written. There are a variety of minor issues that require correction or clarification in the manuscript but most are relatively easily addressed. Clearly this is a rich dataset with a wide number of measures obtained during testing, particularly during gait. It looks like table 2 lists all of the measures examined but it would be helpful for the reader if they were stated more explicitly in the methods, possibly as a list of the planned study measures before they are detailed in the main text. The nonparametric statistical testing for group differences is appropriate, however, it is not clear whether corrections for multiple comparisons were applied to help control for false positive results given the number of tests.

Response from the authors: We would like to thank the reviewer for the valuable comment. We moved the lines 285 to 295 in the original submission to the beginning of the methods section to clarify early in the text which are the main outcomes. We further corrected the level of significance for gait analysis parameters to p < 0.0036 and for the maximum isometric moment to p < 0.0125. The correction had minor effects on the results. Although, the level of significance is low after correction, all the main effects are unchanged and still significant. We hope this is sufficient. 

The Discussion is quite long, which is not necessarily a problem, although there are a number of associations made between pTFA and other populations (e.g. hip luxation, cereberal palsy, Legg Calve’ Perthes, etc.), for which the rationale is not always clear given the inherent differences between such groups.

Response from the authors: We shortened the discussion and deleted the paragraph which referred to different pathologies. 

Page 6 (lines 116-121), The point the authors are trying to make with Sherk et al. is not very clear.

Response from the authors: We would like to thank the reviewer for that comment. We agree that this part was not quite clear. We shortened this part and combined it with the reference to Putz et al. . We hope that this helps for clarification. 

Page 12 (line 249), please clarify why adjustments for the characteristics of the prosthesis were not made when calculating joint kinetics. This would seem likely to have an impact on the measures.

Response from the authors: Thank you for that comment. Adjustments would mainly influence results of joint kinetics during swing phase. We added information that we only investigated kinetics during stance and that correction for the persons/prosthesis characteristics would have a negligible influence onto joint kinetics during stance. This was added in the methods section, at the end of the subsection gait analysis. 

Page 13 (line 266), references Figure 1C which is not present in Figure 1.

Response from the authors: We apologise for that. This was a mistake and supposes to say “B.” we corrected this. Thank you for making us aware of this mistake. 

(line 271), notes that marker trajectories were measured during isometric testing but their use is not clarified later on.

Response from the authors: We would like to thank the reviewer for that comment. We explained the use of the trajectories and the biomechanical model behind OpTIMo in the following paragraph. We used marker trajectories on the pelvis to determine the hip joint centre, similar to CGA methods. Markers on the strap and on the thigh were used to determine the lever and force orientation. This used to be part of the subsection “Data Analysis” in the original submission, which we included in subsection “Isometric muscle strength measurements (OpTIMo)”. We hope this is sufficient. 

(line 287) indicates that the “best trial” was used. Please clarify how this was defined.

Response from the authors: It was the trial with the highest isometric moment (MIM). We added this information in the methods section at the beginning, where we also summarized all main outcome parameters. 

Figure 3, The content of the figure is not clear and the caption does not explain all elements of what is being presented. For example, the titles suggest an x vs. y comparison of the title variables in the plots but the independent variable appears to be gait cycle (although abbreviations are not defined). Shading and dotted lines are not defined in the caption. Please also doublecheck the phrasing ‘none continual scalar’.

Response from the authors: We agree with the reviewer and changed the figure. We hope it is clearer now. We also added more information in the caption for clarification. 

Page 19, reports the spearman correlations between various metrics in TFA but it is unclear how the levels of correlation relate to controls, which are not presented.

Response from the authors: We added correlations of the REF group in the supplement. We added also scatterplots to substantiate the results. After excluding participant no 8 from the analysis we noted significant correlations first found did not persist. Therefore there have been some changes in the discussion where we discuss the relation more moderately. Also, high correlation coefficients in REF group may suggest that there are more relationships (e.g. abd. MIM vs. abd. CGA -0.649 p = 0.004) However, we are critical, as the scatterplots do not suggest clearly that there is a clinical relevant relation in that group(see the plots below). We hope this answers all raised questions of the reviewer.

Page 22 (lines 425-427), if the participants maximum isometric joint moment was being measured in the study (which generally speaking should be the largest moment that can be generated), it is unclear how/why moments generated during gait would necessarily exceed it.

Response from the authors: We would like to thank the reviewer for the comment. We added information in the manuscript that the relationship between the isometric moments and the moments during gait is not straight forward and still subject to research. Fosang et al. also found that dynamic moments exceeded isometric moments, particularly in the ankle joint. We changed parts of the discussion, added the results of Fosang and weakened the conclusions of the first version of the manuscript. We hope this is sufficient. 

Figure S1 caption, please indicate what the ranges correspond to.

Response from the authors: We added the information and also modified the figure slightly to show which gait phases we refer to. We hope this is sufficient for clarification. 

Other minor comments and suggested edits (insertions/deletions in brackets),

Please verify that past tense grammar is consistent throughout the manuscript

Page 4 (line 84), “… might [be] helpful…”

Response from the authors: Changed 

Page 6 (line 127), “…Additional[ly]…”

Response from the authors: This sentence was changed, deleted during rewriting the manuscript. 

(line 132), “…compromise[d]…”

Response from the authors: This sentence was changed, deleted during rewriting the manuscript. 

(line 135), “…who [was] utilizing…”

Response from the authors: This sentence was changed, deleted during rewriting the manuscript. 

Page 7 (line 156), “…walking [is] coronal plane…”

Response from the authors: This sentence was rewritten during rewriting the manuscript. 

(line 159), “…and [] irregularities…”

Response from the authors: Changed 

(line 161), “…TFA [out] of a ….”

Response from the authors: This sentence was rewritten during rewriting the manuscript. 

(line 164), “…this may [also] be related to…”

Response from the authors: Changed 

Page 8 (line 179), “…role of [muscle] strength…”

Response from the authors: The sentence was deleted upon request of the 2nd reviewer, which we agreed to. 

Page 9 (line 190), delete “of the department”

Response from the authors: Changed 

Page 16 (line 325-326), “…vie [of] each side…present[ed]…”, “conjunct” spelling?

Response from the authors: Changed. We replaced conjunct by linked. 

Page 21 (line 390), “…patter[n]…”

Response from the authors: This sentence was rewritten during rewriting the manuscript. 

Page 22 (line 426), “…if this [] working…”

Response from the authors: This sentence was rewritten during rewriting the manuscript. 

Page 23 (line 445), “…socket do[es] not…”

Response from the authors: Changed. 

Reviewer #2: The goal of the manuscript is to examine the influence of hip muscle strength on walking biomechanics for persons with transfemoral amputation. The authors hypothesize that: “people with [transfemoral amputation] TFA will show lower isometric moments on the involved side than those in reference a group (REF); we expect correlations between maximum isometric joint moments and gait kinematics of the hip, pelvis, and trunk; and, further, we expect correlations between maximum hip joint moments in gait with those tested isometrically.” The authors prove their hypothesis and show that persons with transfemoral amputation have reduced hip muscle strength as compared to healthy individuals, but do not attempt to decouple the effects of reduced hip muscle strength and differences in leg length due to the transfemoral prosthesis for the gait deviations observed. It is recommended that the authors perform additional analyses to examine the role of the transfemoral prosthesis on the gait deviations observed rather than ignoring this confounding factor.

Response from the authors: We would like to thank the reviewer for his valuable comments and the summary. We try to respect the reviewer comments and change the manuscript accordingly. We absolutely agree with the reviewer that a leg length discrepancy could be as well one of the cofounding factors which cause gait deviations in people with TFA. We did add the influence of leg length at several points in the manuscript. 

Major:

In the discussion and conclusion, the authors link decreased hip ab/adduction moment to changes in hip and trunk kinematics. While this may be a factor, the reviewer does not believe it is the cause of the kinematic changes. Most prosthetists make create a transfemoral prosthetic system that causes the prosthetic limb to be longer than the sounds limb so that during walking the user compresses the prosthetic foot during stance. This can lead poor hip alignment during standing and during the stance phase of walking if the user does not meaningfully progress onto the prosthesis during this time. Additionally, this longer prosthetic leg can lead to hip hiking during swing. Did the authors examine the trunk obliquity and hip alignment during a static trial? Additionally, the authors could examine the effective leg-length throughout walking by examining a vector from the hip joint center to the center-of-pressure under the foot. Such a vector should be compared to the sound limb to see if the effective limb length is the same. This would allow the authors to make a more concrete statement about how the subjects interact with the prosthesis and if the observed hip and trunk kinematic changes are due to leg length changes or due to weak hip musculature.

Response from the authors: 

We would like to thank the reviewer for the comments. We absolutely agree with the reviewer that a leg length discrepancy will certainly influence whole body movement, including the trunk. We also named this very briefly at the end of the discussion in the previous version of the manuscript (Line 485 in the original submission: “A leg-length discrepancy could also lead to an increased lateral trunk lean.”). To emphasise that trunk movement is one of the possible causes for later trunk displacement we rewrote parts of the introduction and discussion to underline that a leg length discrepancy could be one of these factors. Further we tried to clarify how prosthetic alignment is performed in our facility. We aim for a levelled pelvis during steady standing, with both limbs equally loaded. We also respect the reviewers believe that strength may not be the main influence of lateral trunk movement in people with TFA. However, in this study the emphasis was on strength and the influence onto gait. It is a common belief that strength, in particular hip abductor strength is the only cause for lateral trunk movements in people with TFA. This somehow biased belief is also one reason we conducted this study, as this theory was never proved. We found out that there may be some connection, but we could not deliver true evidence. This underlines the “multifactorial” causes of gait deviations in people with TFA, of which one factor is also the leg length. Other calculations, as suggested by the reviewer will be subject of future research. We would like to thank the reviewer for the inspiration and hope that this sufficiently addresses the critique of the reviewer. 

The introduction and discussion are clunky. The authors often introduce a source and then explain what the study concluded (or the inverse). For example:

“Pröbsting et al. gathered studies on changes in the musculoskeletal system as a result of lower-limb amputation [1]. They categorized these musculoskeletal changes as follows: low back pain, osteoarthritis, reduced bone density, and muscle atrophy.”

This type of referencing occurs 15+ times throughout the introduction and reduces the readability of the article. It is encouraged for the authors to concisely revise the introduction and discussion. An example for the previous quote would be:

“Noted musculoskeletal comorbidities following lower-limb amputation are: low back pain, osteoarthritis, reduced bone density, and muscle atrophy [1].”

Response from the authors: We agree with the reviewer and would like to thank him for making us aware of these shortcomings. We edited and shortened the introduction and the discussion and hope the readability is improved. 

Make sure to check the grammar and punctuation throughout the manuscript. The reviewer has listed several grammar and punctuation items that need to be addressed in the introduction but did not do so for the entire manuscript.

Response from the authors: We would like to thank the reviewer for his efforts. A native speaker and fellow researcher edited the manuscript and we tried to make sure to avoid mistakes and hope we covered all issues. 

Minor:

This is not a prospective study as there was not an extended period of time that the subjects were examined over. Please remove such wording from the title and the manuscript.

Response from the authors: Changed 

Make sure to have continuous past tense throughout the manuscript

Response from the authors: We involved a native speaker for editing and hope we found all shortcomings in language 

Introduction:

Line 91: need a comma instead of and between “integrity” and “perception”

Response from the authors: Changed

Line 102: “this” to “muscles strength” as in the previous sentence you refer to muscle atrophy, amputation and strength

Response from the authors: This sentence was modified during the rewriting the manuscript. 

Line 109: “Medial” instead of ventral? The reviewer does not know what the ventral side of the thigh is.

Response from the authors: Ventral refers to the front. In this case, it refers to the thigh muscles in the front. We changed ventral to anterior, as this may be more commonly used. 

Line 114: add “muscles” to the end of the sentence.

Response from the authors: This sentence was modified during the rewriting the manuscript. 

Line 127: “Addition” to “Additionally,”

Response from the authors: This sentence was modified during the rewriting the manuscript. 

Line 132: “compromise” to “compromised”

Response from the authors: This sentence was modified during the rewriting the manuscript. 

Line 138: “Additionally”

Response from the authors: This sentence was deleted during rewriting the manuscript. 

Line 156-157: “A well-known example of how strength deficits impact walking are probably coronal plane gait deviations of the trunk, as described by Trendelenburg [23].” Is not a definitive sentence.

Response from the authors: We agree with the reviewer and the sentence was rewritten 

Line 178-179: Remove: “We are confident that by using this approach we will gain better insight into the role of strength in people with TFA and their compensation mechanisms in gait.”

Response from the authors: We agree. The sentence was deleted. 

Methods:

How much does the K2 subject skew the results? Should he/she be eliminated from subsequent analyses? This subject, as reported, does not spend as much time on the prosthesis as the other subjects and is the only subject collected that was classified as K2.

Response from the authors: Following this comment and a comment of another reviewer, the influence of this particular subject was questioned. The subject had a lower functional level and used crutches while walking outside. It was consequently decided to remove the subject from the group to recalculate results. 

Lines 190-191: remove “of the department”

Response from the authors: It is removed 

Line 217: Define the acronym MRC

Response from the authors: We defined the acronym MRC (Medical Research Council) 

Lines 221-222: Were these sockets different than their daily use sockets? If they were, how much do you think this affects the gait outcomes of the manuscript?

Response from the authors: The participants used their habitual sockets, which they obtained in our in-house prosthetics and orthotics department. We added this information in the methods. 

Line 225: Define the acronym LASAR

Response from the authors: LASAR stand for “Laser Assisted Static Alignment Reference”. This information was added ion the manuscript. Further we added the reference of Blumentritt et al. who developed this system and defined the acronym. 

Lines 249-250: How does not adjusting for the prosthetic inertial parameters affect the joint kinetics? And how does this affect the comparisons the authors make between able-bodied and persons with transfemoral amputation?

Response from the authors: This was also a request by the 1st reviewer. We added information which underline that such adjustments lead to negligible differences in the calculated joint kinetics, which was confirmed by Pàmies-Vilà et al. [1]. 

Line 266: a comma is needed

Response from the authors: added 

Line 288-291: The reviewer does not understand the how this portion is relevant. What was being correlated to provide a high correlation?

Response from the authors: This refers to the methodology paper which describes the OpTIMo Method in detail [2]. We detected a good reproducibility in this paper with the poorest ICC being 0.765 for hip abduction for the day-to-day variability. As we refer to the methodology paper earlier, we decided to delete this sentence. 

Line 289: put (ICC) after interclass correlation coefficient

Response from the authors: We decided to delete this sentence. 

Line 294: Please define PTFA acronym.

Response from the authors: We apologize for that mistake. We used the acronym PTFA (people with a trans-femoral amputation) in an earlier version of the manuscript. The acronym was changed in the course of the internal review and this is just a “left-over”. We deleted it in this sentence and refer to people with TFA. Thank you for making us aware of this shortcoming. 

Results:

Why compare peak-to-peak maximum isometric moments? This would seem to penalize persons with amputation or hip musculature weakness more. Additionally, this does not provide an accurate characterization of the musculature which is only active in one direction. It is strongly recommended to separate out different motions as is reported in Lines 352 to 355 of the discussion and to not use the peak-to-peak range.

Response from the authors: We agree with the reviewer and do not report peak to peak values any more. We changed the paragraph accordingly. 

Line 313: include “during walking” to the end of the topic sentence.

Response from the authors: Added 

Table 2: It is encouraged to not statistically compare gait parameters between the persons with amputation and healthy individuals as they had a significantly different gait velocity. It has been shown that differences in gait velocity cause differences in lower limb kinematics and kinetics (see Zelik and Kuo 2010).

Response from the authors: We agree with the reviewer. The differences in speed are noticeable and may be of clinical relevance. We added information that there is a noticeable difference in walking speed in the results and in the limitations (at the end of the discussion).

Figure 3: Please update the healthy line color as the reviewer cannot see the mean line.

Response from the authors: We changed the graphs accordingly. 

Figure 3 caption: please include the mean walking velocities for the persons with amputation and healthy individuals.

Response from the authors: We added this information. 

Line 322: this was not a significant finding. why report without significance?

Response from the authors: We removed this sentence as results are shown in the table. 

Can the authors include a table with the variables correlated, the correlation values and the significance of the correlation values?

Response from the authors: Yes, we can and we added a table with the correlation in the supplement. 

Discussion:

Line 350: “Individuals”

Response from the authors: Changed 

Lines 355-358: The reviewer does not see the relevance in reporting the ranges in the MIM. See comment above.

Response from the authors: The sentences which reported ranges were deleted. 

Line 363: “und” to “and”

Response from the authors: We do apologize for that mistake. Changed. 

Lines 368 to 438: Paragraph may want to be broken up to improve readability. Also, the authors refer to many different clinical populations (ex: children with Legg Calve´ Perthes disease, persons with cerebral palsy). This provides un-useful information as many clinical populations may suffer from different co-morbidities which makes the population unique and less relatable to each other.

Response from the authors: We agree with the reviewer and have deleted the parts referring to trunk deviations during walking in other pathologies than people with a lower limb amputation. 

Lines 411-420: Seems to be reiteration of the results with no real discussion going on. Additionally, are all the correlation values significant that are cited?

Response from the authors: We tried to shorten this paragraph, to avoid repetition. We added the p-values to the correlation coefficient. 

Line 425: “In theory one could imagine these peak to peak values as a working range and if this this working range will be surpassed by moments occurring during gait this may induce gait deviations or even make it impossible to walk.” These peak to peak ranges are exceeded during human running (see Riddick et al. 2018) for healthy individuals. The reviewer does not feel that this hypothesis can be supported as human locomotion exceeds these thresholds. It is recommended to temper the language.

Response from the authors: We agree with the reviewer and tampered the language as this is a theory which cannot be proven by the collected data. Unfortunately we could not find the reference the reviewer is citing (maybe the Reviewer is referring to “Soft tissues store and return mechanical energy in human running” Riddick, Kuo 2016, but we are not certain). But we added aspects of the publication of Fosang et al. who also described that moments during walking are surpassing MIM in some instances, i.e. the ankle moment during walking. 

The review thinks that the Author’s statement of: “Strength deficits may force individuals with TFA into compensation mechanisms, which should be distinguished from socket or prosthesis-induced gait alterations. External factors inducing the significant strength deficits in people with TFA cannot be fully identified in this study.” is the crux of the article and should be stated earlier in the discussion than line 462.

Response from the authors: We agree with the reviewer and we state this earlier in the discussion now. 

Line 466-467: “The hip abductors are not directly weakened by the transfemoral amputation itself.” is a misleading statement. The hip abduction muscles may not loose muscle mass during the amputation BUT the thigh muscles are “tacked down” onto the femur or may not even be attached. In the former case, the muscle loses the moment arm at which it acts, effectively making it weaker to perform the necessary task.

Response from the authors: We would like to thank the reviewer for that comment. Unfortunately we do not have information on the amputation technique performed in our cohort, as this could be another cofounding factor. For example, if a myodesis (tacking down the muscle to the bone) or a myoplasty (suture agonists and antagonists) was performed. We changed this sentence including the moment arm and the changed ratio of agonists and antagonists in a TFA. We hope this is clearer now. 

Line 470-481: Why do the authors interpret non-significant results as significant results?

Response from the authors: We tried to rephrase and reformulate these sentences. We just found it peculiar that all muscle groups of the hip showed significant strength deficits in MIM except the hip extensors and tried to find an explanation for that, as this was the only non-significant MIM. Literature and clinical evidence suggests that the extensors play an important role in walking of people with TFA. This made us thinking whether or not there might be a training effect, due to the changed biomechanics which prevents the extensors form losing as much strength as the other muscle groups. We hope this is clearer now. 

Conclusion:

The authors state that this information may be helpful for rehabilitation. The authors should discuss such implication in the discussion.

Response from the authors: Due to the lack of arguments supporting this hypothesis, we decided to delete the sentence. 

Reviewer #3: This manuscript investigates potential correlations of strength gait deviations, as well as compared a cohort of people with transfemoral amputation to people without amputation. The topic of this article is interesting because if we could see a gait deviation and then address it with rehabilitation (e.g., strength or balance), it would help people with amputation walk with less deviations. I have a series of comments below that reference major and minor concerns, and I have highlighted some of them with line numbers but did not highlight every instance that is repeated throughout the manuscript.

General questions: Is maximal strength important with over ground preferred walking speed, or is it more important to show how much of their overall capacity they need to use to walk compared with non-amputees? Is there a greater reserve strength for non-amputees? Maybe a deviation is chosen to save available muscle fibers and reserve capacity to respond to perturbations or minimize energy expenditure.

Response from the authors: we would like to thank the reviewer for these remarks.

Response to the general questions above: For normal walking in unimpaired people somebody will most likely not need their maximum effort. So, one of our aims was to prove the second question: “how much of their overall capacity they need to use to walk compared with non-amputees” Is there a greater reserve strength for non-amputees? Yes, we were able to show this, e.g. in Figure 2 the horizontal lines representing the MIM in the according direction can be understood as a “Working Range” or “Reserve”. 

“Maybe a deviation is chosen to save available muscle fibers and reserve capacity to respond to perturbations or minimize energy expenditure? „We have a similar concept. See the sentences in the discussion: “The strength status of people with TFA is highly clinically relevant. Strength deficits may force individuals with TFA into compensation mechanisms during walking, which should be distinguished from socket-, prosthetic-component, or prosthesis-leg length induced gait alterations.”

We hope the changes and the revised manuscript help to clarify all raised questions. 

Major Comments

Abstract:

Line 51: Is this a prospective study? A prospective study identifies a population and then follows them and tries to make predictions about what will happen, whereas here you are trying to make correlations between gait deviations and strength deficits.

Response from the authors: We agree with the reviewer and deleted the parts which refer to a prospective study. 

Lines 51-53: Aside from the prospective study comment, I like this statement because it is clear about what you did in this study.

Response from the authors: We would like to thank the reviewer for the encouraging comment and we deleted the prospective study comment. 

Introduction:

Line 95: This is something done many times throughout the manuscript, and may just be a personal preference on my part, but other authors should be cited while their findings are prominent. “Probsting…” sentence adds nothing. Then line 104, “Ostchega et al….” did not prove a relationship, the results or findings of a study show something, the authors just write it up. Also, there are very few proven relationships, there is more evidence for some things than others, but one study (and multiple studies) does not prove a relationship.

Response from the authors: We would like to thank the reviewer for this remark and follow the suggestions in the revised version of the manuscript. 

Line 102: Unless you plan on talking about other forms of mobility, such as transfers, just say “this relates to walking.”

Response from the authors: Changed 

Throughout the introduction there could be more standalone paragraphs, and rather than listing findings from study after study, the information could be synthesized or summarized. This relates to my comment above about authors not showing anything, but the overall data collected from many studies.

Response from the authors: We tried to rewrite larger parts of the introduction as well of the discussion. We tried to synthesize the statements from other studies and bring them better in the context of our results. Lastly we tried to structure the introduction in several thematic paragraphs. We hope the critique will be milder in the revised manuscript. Although we have to admit that we cited many studies, as we wanted to draw a picture of the many factors which could lead to gait deviations. 

Lines 173 and 174: There should not be methods in the introduction.

Response from the authors: We agree with the authors and deleted the part referring explicitly to the method. 

Lines 182-183: Should hypothesized directionality of correlations, not just state there will be correlations.

Response from the authors: We agree with the reviewer and tried to rephrase the sentence and describe in which directions we expect relationships. 

Methods:

Line 213-215: I am not sure if participant 8 should be included in the data analysis, especially since he is a K2 ambulator and then he walked without crutches during his motion analysis. Walking without crutches is not his normal or preferred, and therefore the gait speed and mechanics are not preferred, whereas the participants were using preferred mechanics.

Response from the authors: We agree with the reviewer that this participant differs from the other participant. Therefore, we excluded patient no8 from the analysis and recalculated the results.

Lines 267-268: Did you standardize the amount of test trials for participants to try the OpTIMo?

Response from the authors: Yes we did. Each participant performed two repetitions for each motion direction with maximal force. Number of trials for getting familiar with the system, with submaximal force was not standardised. We added this information in the methods section. 

Lines 284: General questions about OpTIMo, how widespread is this device clinically or in research? Is it a research-only device? How does it compare to typical clinical force dynamometry measures with patients laying down?

Response from the authors: We would like to thank the reviewer for these questions and feel complimented for the interest in this device. 1st: The OpTIMo device is a development of the local laboratory and is not used elsewhere, to our knowledge. However there are other research group combining force-transducers or hand held dynamometers with optical marker data to obtain joint centres, length and orientation of the lever to calculate joint moments. 2nd: OpTIMo is a research only device and was specifically developed to collect strength data during conventional gait analysis. 3rd: We have not compared results of the device with other strength measurement methods. The idea is that we determine hip joint centres in a similar way as in conventional gait analysis, therefore marker on the pelvis have to be visible, which would not be possible with a person in supine position. Further we specifically wanted the people to stand upright during the test, as we believe that this position is closer to the function of walking, as we are aiming for a comparison of walking and isometric strength. 

Line 287: What makes a trial the best?

Response from the authors: We added the requested information. It was the trial with the highest MIM. 

Line 295: Why not look at the sound side too? Could be interesting when comparing to affected side and unaffected participants.

Response from the authors: In this study we aimed for the involved side as this clearly has a changed muscle configuration. We would like to thank the reviewer for that comment and the inspiration. This comparison was not in the scope of our study, but sound side measurements could potentially be performed in the future. 

Statistics look right.

Response from the authors: As there were a few legitimate questions raised by the other reviewers we changed statistics according to their requests.

Results:

Line 310-311: All the differences were significant except for the step length.

Response from the authors: We do apologise for this avoidable mistake and changed it accordingly. 

Figure 3: Should label x and y axes rather than leave that to the title. Also, I do not know what FS, FO, oFS, and oFO are, I assume they are foot strike and foot off, but do not know.

Response from the authors: We changed to figure for clarification as we realised that the term “vs.” may cause the reader to interpret the graphs as a plot of one value vs. the other. In this case it is a time series normalized to the gait cycle with the scalar, isometric moments as vertical lines for reference. 

Lines 342-345: What does negative and positive correlations mean in your results?

Response from the authors: It depends on the data. In trunk motion, higher value means greater gait deviation (i.e. we expect a reduction of trunk movement, with greater abduction MIM). In case of MIM we obtained a positive correlation, as greater values in one MIM will lead to greater values in another MIM. The sign is somehow secondary, as it is not really helpful to determine whether or not there is a correlation it merely gives an insight if we are looking at a rising or falling slope. 

Discussion:

General comment: Should not repeat results, especially with p-values, in the discussion section, they should be in the results section alone.

Response from the authors: Deleted 

Line 375: Remove distinguished, that is your spin and unnecessary.

Response from the authors: Deleted 

Line 382: The results from Cappozzo agree with your results, do not confirm your results, and if anything, your results confirm their finding.

Response from the authors: Changed 

Lines 409-421: Listing many results in the discussion section, could be synthesized with the literature better.

Response from the authors: We have rewritten larger parts of the discussion in order to synthesis literature references better. 

Minor Comments

Abstract:

Line 47: Awkward wording “on the person concerned”

Response from the authors: Deleted 

Line 59: should be “Kinematics” not “Kinematic”

Response from the authors: Changed 

Line 59: Could say determined, not derived, unless you derived equations.

Response from the authors: Thank you for this comment, we changed it accordingly 

Line 72: Could say “were identified, i.e. significantly lower speed,…”

Response from the authors: Added 

Introduction:

Line 91: Could say “body integrity, perception…”

Response from the authors: Changed 

Line 98: “clear atrophy” does not make sense, atrophy alone works

Response from the authors: This sentence was deleted in the course of rewriting the introduction. 

Line 98: I recommend saying amputated or affected side, rather than ipsilateral side. I understand ipsilateral but I am not sure every reader will, it is more clear to say something like amputated, affected or involved.

Response from the authors: This sentence was deleted in the course of rewriting the introduction. We further avoided the word ipsilateral throughout the manuscript. 

Line 102: “particularly” does not add anything, like most adverbs

Response from the authors: Deleted 

Line 126: What do you mean by “redundant”?

Response from the authors: We agree with the reviewer that redundant was not precise and changed it to superfluous 

Line 127: Could say “Additionally, the…”

Response from the authors: Changed 

Line 128: What do you mean by “finite coupling”?

Response from the authors: We changed it to “somewhat stiff coupling”

Line 130: Could say “Gholizadeh et al. underline the problems in…”

Response from the authors: Changed 

Line 138: Could say “Additionally, new…”

Response from the authors: Changed 

Line 164: Lower case i in ischial

Response from the authors: This sentence was deleted. 

Methods:

Line 193: clinical gait analysis can be replaced with CGA because the acronym is already listed in the beginning

Response from the authors: Apologies. We changed this accordingly. 

Line 225: Is the L.A.S.A.R. posture device widely known? I am not sure if it should be explained more or you could stop the sentence with “recommendations.”

Response from the authors: The L.A.S.A.R. posture device is widely known. We added a very brief description and a reference. So, the interested reader has the possibility to get additional information on the device. 

Line 240: What was the collection rate for the force plates?

Response from the authors: We were not sure if the reviewer is referring to the sampling rate or the number of trials capured to calculate a mean. We added both in the methods section. 

Results:

Line 315: “in the patients” is awkward wording

Response from the authors: We changed this sentence. 

Lines 320-321: “For CGA…the patient group.” is not a clear sentence, could be written better.

Response from the authors: We totally agree with the reviewer and changed the sentence to: “During walking, the participants with TFA showed significantly lower peak internal hip abduction and adduction moments in stance, as well as peak hip extension moment in stance when compared to REF.” >> We hope this improves readability and is clearer. 

Line 325: Why is this schematic “exemplary”?

Response from the authors: We removed the word exemplary and added the information that the schematics show a representative trial. 

In table 2, Kg should be kg, and it should be N m and not Nm because they are 2 separate words.

Response from the authors: Apologies for overseeing these mistakes. We thank the reviewer for making us aware of it. We changed it. Also Nm in the figures were changed to N m. 

Line 339: Should say “with TFA hip MIM of…” and not “with TFA Hip MIMof…”

Response from the authors: Changed 

Discussion:

Line 390: pattern, not patter

Response from the authors: Changed 

Line 445: Say For example, no E.g. to lead a sentence

Response from the authors: Changed! Thank you, we were not aware of this rule. 

Line 445: does rather than do

Response from the authors: Changed 

Line 449-452: Not a clear sentence.

Response from the authors: This sentence was deleted in the course of rewriting the manuscript. 

Line 452: stabilize rather than stabile

Response from the authors: This sentence was deleted in the course of rewriting the manuscript. 

Line 501: address rather than tackle deficits, tackle is in American football or Rugby rather deficits of our patients/clients 

Response from the authors: This sentence was deleted in the course of rewriting the manuscript. 

1. Pàmies-Vilà R, Font-Llagunes JM, Cuadrado J, Alonso FJ. Analysis of different uncertainties in the inverse dynamic analysis of human gait. Mechanism and Machine Theory. 2012;58:153-64. doi: https://doi.org/10.1016/j.mechmachtheory.2012.07.010.

2. Heitzmann DW, Guenther M, Becher B, Alimusaj M, Block J, van Drongelen S, et al. Integrating strength tests of amputees within the protocol of conventional clinical gait analysis: a novel approach. Biomedizinische Technik Biomedical engineering. 2013;58(2):195-204. doi: 10.1515/bmt-2012-0036. PubMed PMID: 23454713.

---

## [Decision Letter · Decision Letter 1]

3 Jul 2020

PONE-D-19-24706R1

The influence of hip muscle strength on gait in individuals with a unilateral transfemoral amputation

PLOS ONE

Dear Dr. Heitzmann,

Thank you for submitting your manuscript to PLOS ONE. After careful consideration, we feel that it has merit but does not fully meet PLOS ONE’s publication criteria as it currently stands. Therefore, we invite you to submit a revised version of the manuscript that addresses the points raised during the review process.

We look forward to receiving your revised manuscript.

Kind regards,

Yih-Kuen Jan, PhD

Academic Editor

PLOS ONE

Reviewers' comments:

Reviewer's Responses to Questions

**Comments to the Author**

1. If the authors have adequately addressed your comments raised in a previous round of review and you feel that this manuscript is now acceptable for publication, you may indicate that here to bypass the “Comments to the Author” section, enter your conflict of interest statement in the “Confidential to Editor” section, and submit your "Accept" recommendation.

Reviewer #1: (No Response)

Reviewer #2: (No Response)

Reviewer #3: All comments have been addressed

2. Is the manuscript technically sound, and do the data support the conclusions?

Reviewer #1: Yes

Reviewer #2: Yes

Reviewer #3: Yes

3. Has the statistical analysis been performed appropriately and rigorously? 

Reviewer #1: Yes

Reviewer #2: Yes

Reviewer #3: Yes

4. Have the authors made all data underlying the findings in their manuscript fully available?

Reviewer #1: Yes

Reviewer #2: Yes

Reviewer #3: Yes

5. Is the manuscript presented in an intelligible fashion and written in standard English?

Reviewer #1: Yes

Reviewer #2: Yes

Reviewer #3: Yes

6. Review Comments to the Author

Reviewer #1: The revised manuscript has been extensively rewritten in response to the reviewers’ comments. The literature in the introduction is now presented in a more synthesized form, the statistical analyses have been revised to account for multiple comparisons, and the discussion has largely been re-written to better relate the results to the literature and discuss the impact of potential confounds. In doing so, the clarity of the manuscript has been improved. A few minor issues remain, which should be addressed prior to publication.

The methods indicate that Spearman’s rank correlation was used, but in some places the results refer to “linear correlations” (e.g., Line 505 and S2_Figure). If the references to a linear correlation are correct, please include in the methods. In not, I would suggest removing the reference to ‘linear’ and the line fits in S2_Figure, since the spearman rank correlation characterizes the monotonic relationship between two variables, which is less restrictive than assuming a linear relationship.

Lines 278-280, suggest revising to read, “The level of significance for correlations was corrected to p < .0036 to account for testing across the coronal and sagittal plane CGA and MIM parameters.”

Lines 293 & 297, I believe the table being referenced is Table 2.

Line 336, the labels (and definitions) for FS, oFO, oFS, and FO are not in the revised figure and can be removed.

Line 376 references S2_Fig when stating that “…pelvic drop of the none-involved side was not present in our TFA group…”. Please clarify how S2_Fig shows this.

Lines 451-0452, suggest revising to read, “For example, the affected side ground contact in a person with TFA is different than in an unimpaired person.”

Lines 454-455, the phrase “adapt the introduced”, seems incomplete. Suggest revising.

Lines 455-457, suggest revising to read, “Another indicator that coronal plane gait deviations cannot purely be attributed to hip strength deficits is that a lateral trunk lean can also present in amputations below the knee.”

Lines 460-461, the phrase “…the reasons behind the strength deficits and the commonly, to strength deficits attributed gait deviations…” seems incomplete. Suggest revising.

Line 468, suggest revising to read, “may induce a weakening of certain muscle groups…”

Line 491, suggest revising to read, “participants reported being satisfied with…”

Lines 507-509 state that “Over the course of this study, an additional hypothesis was developed, i.e. that people with TFA should not have a maximum hip moment below a certain threshold to avoid gait deviations.”. There does not appear to be any indication of a threshold effect in the results. Please clarify how the current data/analyses directly test this hypothesis.

Lines 515-518, suggest revising to read, “However, during walking compensation mechanisms may also be induced, e.g. by the socket, prosthetic leg length or prosthetic component related, that contribute to the net functional deficit.”

Other minor comments and suggested edits (insertions/deletions in brackets),

Line 80, “…reduced bone density and also [] volume loss…”

Line 97, “Larger amounts of fat embedded in the residual limb[] muscles…”

Line 174, “…aids differ[red] considerably from…”

Line 290, “For temporal spatial parameters[,] individuals…”

Line 294, “For CGA kinematic results[,] participants with…”

Line 339, “…extension, and flexion) in [] both…”

Line 340, “For hip abduction and adduction[,] MIMs in individuals…”

Line 372-373, “However, the patter[n] described by…”

Line 376, “…pelvic drop of the non[]-involved side…”

Line 378, “…described a different patter[n] in a different pathology…”

Line 381, “…with TFA show[,] in contrast to Trendelenburg[,] a coronal plane…”

Line 389, “Although not clearly detailed[,] this muscle…”

Line 415, “…individuals with TFA)[,] we observed…”

Line 443, “…it may be advisable to [guarantee] passive

Line 480, “…has negative impact on[] the preferred…”

Line 481, “Consequentially, there seem[s] to be…”

Reviewer #2: First off, I would like to thank the authors for taking time to improve their manuscript. This has really shown through in the introduction which has a better flow than in the previous version. My comments are mostly dedicated to updating the phrasing throughout the manuscript and restructuring of the discussion to make the manuscript more concise and link similar ideas together.

Minor:

Abstract:

Line 37: please update “convenience” to “convenient”

Line 45: please update “abduction, adduction[,] extension[,] and flexion” with the commas suggested

Line 56: consider updating “with significantly higher ranges of motion on the involved side” to “with significantly higher ranges of motion during involved side stance phase”. I suggest adding “stance phase” to the sentence as the trunk is not a part of the involved side.

Introduction:

Line 81: Delete “, as described by Probsting et al.” as this is implicit with the citation.

Line 87: Change “conversely” to “furthermore” as the authors are adding to the argument established in the previous sentence.

Lines 97-99: Consider deleting the last sentence of this paragraph as it distracts from the points that the author has established about muscle strength and atrophy.

Lines 107-112: Consider updating: “In contrast to this, a more rigid fixation of the residual limb with the prosthesis, i.e. by osseointegration, or bone anchored prostheses, in individuals with TFA may have a positive effect on muscle strength. This hypothesis was supported in a study by Leijendekkers et al. who measured muscle volume. In the monitored subject with TFA the hip abductor muscle volume increased in the sound limb by 5.5% and in the residual limb by 7.4% during 12 months after implanting the bone-anchored prosthesis [16].”

to:

“In contrast to this, a more rigid fixation of the residual limb with the prosthesis, i.e. by osseointegration, or bone anchored prostheses, in individuals with TFA can have a positive effect on muscle strength [16].”

Methods:

The methods overall read nicely.

Page 11: Thank you for including more information regarding you TFA population and the procedure for prosthetic/socket fit.

Line 219: Consider deleting “following Kadaba et al. and Davis et al.” as this information is included in the citations.

Line 224: “was” to “were”

Line 230: consider adding the following commas to the sentence: “The OpTIMo device[,] used to determine isometric hip moment[s][,] consist of a rigid frame….”

Lines 278-279: consider deleting “as well” as it makes the sentence clunky

Results:

The results read nicely.

Line 290: add a comma between “parameters” and “individuals”

Line 294: add a comma between “results” and “participants”

Discussion:

The following are recommendation to make the discussion more concise and to bring together similar ideas that were throughout the discussion.

Line 356: consider deleting “TTA” and updating the sentence to “persons with TFA” because it does not make sense to connect the results with persons with TTA as they are a completely different population.

Lines 357-361: consider deleting these two sentences as it has you have already summarized these results in the previous sentence.

Line 362-363: Why do the authors think that there was no strength deficit in hip extension as there were deficits the other directions? The authors have an explanation for this later in the discussion. I would recommend that they move lines 472-482 to the end of this paragraph. Please update some of these sentences as to not use the [author] et al…. (explanation). Updating the sentence structure will make the sentences clear and concise.

Line 367: consider updating “range” to “range-of-motion”

Lines 371-378: Consider deleting these sentences as they do not add meaningful content to the author’s discussion as these sentences are dedicated to a population with a different pathology – which is stated in the last sentence fragment.

Line 369: Consider adding the following to the end of the sentence: “, as has been observed in people with TFA [24,25,42,45]”. The authors can then delete the sentence from lines 379-380.

Lines 383-391: It is suggested not to use the phrasing: [author] et al. showed….. this style of writing makes these sentences difficult to read. This information may be able to be summarized concisely. For example: “Altered trunk and hip kinematics exhibited in persons with TFA have been attributed to weak hip musculature [13,24,42].”

The authors could then move lines 398-403 after this statement as it connects very well. In my opinion, the authors should have the caveat in this paragraph that such a correlation was not observed in the healthy population. The S2 Figure (which I like) shows that less strength in the REF group would have a reduced trunk obliquity (p=0.02).

For instance: (the following is a suggestion for the authors, linking these statements together eliminates the reader having to go back and forth between paragraphs separated by another paragraph)

Altered trunk and hip kinematics exhibited in persons with TFA have been attributed to weak hip musculature [13,24,42]. This connection may explain the observed, non-significant, moderate negative correlation between hip abduction MIM and trunk obliquity range-of-motion. This correlation indicates that weaker people with TFA have the tendency to show a greater trunk range-of-motion during walking than their stronger piers. However, it should be noted that this is the exact opposite trend that was observed in the healthy population (S2 Fig.).

This may lead to lines 403-406 may be able to be appended (with some editing) to the paragraph from lines 392-397.

Lines 409-410: consider update the start of the sentence to: “In typically developing children, the hip MIM…”

Line 426: consider updating “contradictable” to “difficult”

Lines 441-443: This sentence is very similar to the previous. It is recommended to end the sentence on line 441 with two citations [44,45] to eliminate the redundancy. I recommend then moving lines 445-447 after this sentence because it directly connects.

For example:

“For example, it is well accepted that quadrilateral sockets do not stabilize the femur in the coronal plane and therefore will more likely cause increased residual limb movement during walking, compared to other designs, e.g. ischial containment sockets [44, 45]. Decreased medio-lateral stability residuum and the femur, as a cause for a lateral trunk motion, along with other cofounding factors, like strength deficits of the hip abductors the socket [46]. The strength status of people with TFA is therefore of high clinical relevance.”

This recommendation does remove “So, in socket design it may be advisable to grantee passive stability and a very good force transmission, while not restricting the remaining muscles.” however, the proposed change does capture the importance of socket design.

499: “K_level” to “K-level”

Conclusion:

Looks good!

Reviewer #3: The authors made significant improvements to the manuscript. Overall, the writing is better and the authors have addressed my and other reviewer's comments. There a few minor issues to revise before being ready for publication.

Slight formatting issues with table 1, column width is too narrow.

Line 28: I am assuming you mean mass, not weight.

Line 443: grantee, incorrect word?

Repeated use of consequentially, a different word or phrase in few instances to improve readability

Check for consistent spacing after periods, fluctuates between 0 to 2 spaces

Check - versus _ in regards to K-levels (e.g., lines 498-499)

Thank you for addressing our comments.

7. PLOS authors have the option to publish the peer review history of their article (what does this mean?). If published, this will include your full peer review and any attached files.

Reviewer #1: **Yes: **Scott A. Beardsley

Reviewer #2: No

Reviewer #3: No

---

## [Author Response · Author response to Decision Letter 1]

4 Jul 2020

Reviewer #1: The revised manuscript has been extensively rewritten in response to the reviewers’ comments. The literature in the introduction is now presented in a more synthesized form, the statistical analyses have been revised to account for multiple comparisons, and the discussion has largely been re-written to better relate the results to the literature and discuss the impact of potential confounds. In doing so, the clarity of the manuscript has been improved. A few minor issues remain, which should be addressed prior to publication.

Response to the Reviewer: We would like to thank the reviewer for the encouraging words and the many valuable suggestions. This was more than helpful and we appreciate the time and effort the reviewer put into their response.

The methods indicate that Spearman’s rank correlation was used, but in some places the results refer to “linear correlations” (e.g., Line 505 and S2_Figure). If the references to a linear correlation are correct, please include in the methods. In not, I would suggest removing the reference to ‘linear’ and the line fits in S2_Figure, since the spearman rank correlation characterizes the monotonic relationship between two variables, which is less restrictive than assuming a linear relationship.

Response to the Reviewer: We would like to thank the reviewer for the very valuable input. The reviewer is correct; we used Spearman’s rank correlation. So, we deleted passages which referred to a “linear” behaviour. As the reviewer is also questioning the hypothesis formulated in this paragraph, we decided to delete it completely. We agree with the reviewer that this hypothesis was a bit too motivated and was partly based on the results including one participant, which we excluded in the course of the previous review. We further deleted the lines in S2_Fig. which did suggest that we tested for a linear relationship. 

Lines 278-280, suggest revising to read, “The level of significance for correlations was corrected to p < .0036 to account for testing across the coronal and sagittal plane CGA and MIM parameters.”

Response to the reviewer: We would like to thank the reviewer for this suggestion, which we changed accordingly in the new manuscript (now Lines 272-274)

Lines 293 & 297, I believe the table being referenced is Table 2.

Response to the reviewer: Correct! We apologise for this mistake, and changed the reference to table 2.

Line 336, the labels (and definitions) for FS, oFO, oFS, and FO are not in the revised figure and can be removed.

Response to the reviewer: The abbreviations for the discrete gait events are still present in the figure. They may not be as obvious as before, because we added the sub-phases underneath. So, we did not change the caption of the figure. 

Line 376 references S2_Fig when stating that “…pelvic drop of the none-involved side was not present in our TFA group…”. Please clarify how S2_Fig shows this.

Response to the reviewer: We would like to thank the reviewer for noticing. This is a mistake and it should say “Fig. 2” not “S2 Fig.”. This part of the discussion was further rewritten.

Lines 451-0452, suggest revising to read, “For example, the affected side ground contact in a person with TFA is different than in an unimpaired person.”

Response to the reviewer: Thank you. We changed this sentence to: “Moreover, the affected side ground contact in a person with TFA is different than in an unimpaired person.” (Now in line 416-417)

Lines 454-455, the phrase “adapt the introduced”, seems incomplete. Suggest revising.

Response to the reviewer: Thank you. We revised this sentence to: “Thus, gait deviations in this population may be a response to unwanted loads introduced to the involved side, and may serve as an option to readjust loads.” (Now in line 419-421)

Lines 455-457, suggest revising to read, “Another indicator that coronal plane gait deviations cannot purely be attributed to hip strength deficits is that a lateral trunk lean can also present in amputations below the knee.”

Response to the reviewer: Thank you. We changed this sentence according to your suggestion (Now in line 421-422).

Lines 460-461, the phrase “…the reasons behind the strength deficits and the commonly, to strength deficits attributed gait deviations…” seems incomplete. Suggest revising.

Response to the reviewer: Thank you. Yes, this sentence was incomplete and we revised it to: “Hence, the origins for strength deficits in people with TFA and their gait deviations, partially attributed to these strength deficits, cannot be fully clarified in this study.” (now in line 425-426)

Line 468, suggest revising to read, “may induce a weakening of certain muscle groups…”

Response to the reviewer: Changed

Line 491, suggest revising to read, “participants reported being satisfied with…”

Response to the reviewer: Changed

Lines 507-509 state that “Over the course of this study, an additional hypothesis was developed, i.e. that people with TFA should not have a maximum hip moment below a certain threshold to avoid gait deviations.”. There does not appear to be any indication of a threshold effect in the results. Please clarify how the current data/analyses directly test this hypothesis.

Response to the reviewer: We deleted this paragraph completely. As stated earlier, we agree with the reviewer that this hypothesis was a bit too motivated and was partly based on the results including one participant, which we excluded in the course of the previous review.

Lines 515-518, suggest revising to read, “However, during walking compensation mechanisms may also be induced, e.g. by the socket, prosthetic leg length or prosthetic component related, that contribute to the net functional deficit.”

Response to the reviewer: Thank you. We changed this sentence according to your suggestion (now in line 470-471)

Other minor comments and suggested edits (insertions/deletions in brackets),

Line 80, “…reduced bone density and also [] volume loss…”

Response to the reviewer: Changed

Line 97, “Larger amounts of fat embedded in the residual limb[] muscles…”

Response to the reviewer: Changed

Line 174, “…aids differ[red] considerably from…”

Response to the reviewer: Changed

Line 290, “For temporal spatial parameters[,] individuals…”

Response to the reviewer: Changed

Line 294, “For CGA kinematic results[,] participants with…”

Response to the reviewer: Changed

Line 339, “…extension, and flexion) in [] both…”

Response to the reviewer: Changed

Line 340, “For hip abduction and adduction[,] MIMs in individuals…”

Response to the reviewer: Changed

Line 372-373, “However, the patter[n] described by…”

Response to the reviewer: Changed

Line 376, “…pelvic drop of the non[]-involved side…”

Response to the reviewer: Changed

Line 378, “…described a different patter[n] in a different pathology…”

Response to the reviewer: Changed

Line 381, “…with TFA show[,] in contrast to Trendelenburg[,] a coronal plane…”

Response to the reviewer: Changed

Line 389, “Although not clearly detailed[,] this muscle…”

Response to the reviewer: Changed

Line 415, “…individuals with TFA)[,] we observed…”

Response to the reviewer: Changed

Line 443, “…it may be advisable to [guarantee] passive

Response to the reviewer: Apologies for overseeing the wrong autocorrect result. We changed this.

Line 480, “…has negative impact on[] the preferred…”

Response to the reviewer: Changed

Line 481, “Consequentially, there seem[s] to be…”

Response to the reviewer: Changed

Reviewer #2: First off, I would like to thank the authors for taking time to improve their manuscript. This has really shown through in the introduction which has a better flow than in the previous version. My comments are mostly dedicated to updating the phrasing throughout the manuscript and restructuring of the discussion to make the manuscript more concise and link similar ideas together.

Response to the reviewer: We would like to thank the reviewer for the inspiring words. We also like to thank the reviewer for invested time and effort. 

Minor:

Abstract:

Line 37: please update “convenience” to “convenient”

Response to the reviewer: Changed

Line 45: please update “abduction, adduction[,] extension[,] and flexion” with the commas suggested

Response to the reviewer: Changed

Line 56: consider updating “with significantly higher ranges of motion on the involved side” to “with significantly higher ranges of motion during involved side stance phase”. I suggest adding “stance phase” to the sentence as the trunk is not a part of the involved side.

Response to the reviewer: We like to thank the reviver for this valuable suggestion. We agree to this remark and changed the sentence accordingly (now in line 54-55)

Introduction:

Line 81: Delete “, as described by Probsting et al.” as this is implicit with the citation.

Response to the reviewer: Deleted

Line 87: Change “conversely” to “furthermore” as the authors are adding to the argument established in the previous sentence.

Response to the reviewer: Changed

Lines 97-99: Consider deleting the last sentence of this paragraph as it distracts from the points that the author has established about muscle strength and atrophy.

Response to the reviewer: We partly disagree with the reviewer and didn’t delete this. We have rewritten the sentences as the fatty degeneration is part of the structural change and muscle atrophy of the residual limb (now in line 94-97).

Lines 107-112: Consider updating: “In contrast to this, a more rigid fixation of the residual limb with the prosthesis, i.e. by osseointegration, or bone anchored prostheses, in individuals with TFA may have a positive effect on muscle strength. This hypothesis was supported in a study by Leijendekkers et al. who measured muscle volume. In the monitored subject with TFA the hip abductor muscle volume increased in the sound limb by 5.5% and in the residual limb by 7.4% during 12 months after implanting the bone-anchored prosthesis [16].”

to:

“In contrast to this, a more rigid fixation of the residual limb with the prosthesis, i.e. by osseointegration, or bone anchored prostheses, in individuals with TFA can have a positive effect on muscle strength [16].”

Response to the reviewer: Thank you for the valuable suggestion which wraps up the information very well. We have changed this accordingly (now in line 104-107).

Methods:

The methods overall read nicely.

Response to the reviewer: Thank you very much! 

Page 11: Thank you for including more information regarding you TFA population and the procedure for prosthetic/socket fit.

Response to the reviewer: We appreciate that the reviewer is recognising that such details are important to us. Thank you!

Line 219: Consider deleting “following Kadaba et al. and Davis et al.” as this information is included in the citations.

Response to the reviewer: Deleted

Line 224: “was” to “were”

Response to the reviewer: Changed

Line 230: consider adding the following commas to the sentence: “The OpTIMo device[,] used to determine isometric hip moment[s][,] consist of a rigid frame….”

Response to the reviewer: Changed

Lines 278-279: consider deleting “as well” as it makes the sentence clunky

Response to the reviewer: Thank you for the suggestion. This shortcoming was also recognized by the 1st reviewer and we changed the sentence according to his suggestion.

Results:

The results read nicely.

Response to the reviewer: Thank you!

Line 290: add a comma between “parameters” and “individuals”

Response to the reviewer: Changed. This was also identified by the 1st reviewer.

Line 294: add a comma between “results” and “participants”

Response to the reviewer: Changed. This was also identified by the 1st reviewer.

Discussion:

The following are recommendation to make the discussion more concise and to bring together similar ideas that were throughout the discussion.

Response to the reviewer: Thank you, we appreciate your suggestions.

Line 356: consider deleting “TTA” and updating the sentence to “persons with TFA” because it does not make sense to connect the results with persons with TTA as they are a completely different population.

Response to the reviewer: Deleted.

Lines 357-361: consider deleting these two sentences as it has you have already summarized these results in the previous sentence.

Response to the reviewer: We agree with the reviewer and deleted the partly redundant information.

Line 362-363: Why do the authors think that there was no strength deficit in hip extension as there were deficits the other directions? The authors have an explanation for this later in the discussion. I would recommend that they move lines 472-482 to the end of this paragraph.

Response to the reviewer: We agree with the reviewer and moved the sentence accordingly.

Please update some of these sentences as to not use the [author] et al…. (explanation). Updating the sentence structure will make the sentences clear and concise.

Response to the reviewer: We tried to delete most of the criticised sentences.

Line 367: consider updating “range” to “range-of-motion”

Response to the reviewer: Changed

Lines 371-378: Consider deleting these sentences as they do not add meaningful content to the author’s discussion as these sentences are dedicated to a population with a different pathology – which is stated in the last sentence fragment.

Response to the reviewer: We partly agree with the reviewer, that this is partly an off-topic. However, the pattern seen in people with TFA is often described as a Trendelenburg gait pattern, in particular in our community. We try to emphasis with these sentences that this is actually not correct. We deleted redundant parts of this section and tried to make it more concise. We hope that the reviewer is accepting our changes (see lines 359 following)

Line 369: Consider adding the following to the end of the sentence: “, as has been observed in people with TFA [24,25,42,45]”. The authors can then delete the sentence from lines 379-380.

Response to the reviewer: Thank you for the suggestion. We changed this accordingly (now in line 356-358)

Lines 383-391: It is suggested not to use the phrasing: [author] et al. showed….. this style of writing makes these sentences difficult to read. This information may be able to be summarized concisely. For example: “Altered trunk and hip kinematics exhibited in persons with TFA have been attributed to weak hip musculature [13,24,42].”

Response to the reviewer: Thank you for the suggestion. We tried to take this into account. The paragraph was changed accordingly (now in line 366-367).

The authors could then move lines 398-403 after this statement as it connects very well. 

Response to the reviewer: We moved the section as suggested.

In my opinion, the authors should have the caveat in this paragraph that such a correlation was not observed in the healthy population. The S2 Figure (which I like) shows that less strength in the REF group would have a reduced trunk obliquity (p=0.02).

For instance: (the following is a suggestion for the authors, linking these statements together eliminates the reader having to go back and forth between paragraphs separated by another paragraph)

Altered trunk and hip kinematics exhibited in persons with TFA have been attributed to weak hip musculature [13,24,42]. This connection may explain the observed, non-significant, moderate negative correlation between hip abduction MIM and trunk obliquity range-of-motion. This correlation indicates that weaker people with TFA have the tendency to show a greater trunk range-of-motion during walking than their stronger piers. However, it should be noted that this is the exact opposite trend that was observed in the healthy population (S2 Fig.).

Response to the reviewer: We absolutely agree with the reviewer changed the section as suggested. This was very helpful and will improve readability. Thank you!

This may lead to lines 403-406 may be able to be appended (with some editing) to the paragraph from lines 392-397.

Response to the reviewer: Changed, shortened and edited accordingly.

Lines 409-410: consider update the start of the sentence to: “In typically developing children, the hip MIM…”

Response to the reviewer: Changed

Line 426: consider updating “contradictable” to “difficult”

Response to the reviewer: Changed

Lines 441-443: This sentence is very similar to the previous. It is recommended to end the sentence on line 441 with two citations [44,45] to eliminate the redundancy. I recommend then moving lines 445-447 after this sentence because it directly connects.

For example:

“For example, it is well accepted that quadrilateral sockets do not stabilize the femur in the coronal plane and therefore will more likely cause increased residual limb movement during walking, compared to other designs, e.g. ischial containment sockets [44, 45]. Decreased medio-lateral stability residuum and the femur, as a cause for a lateral trunk motion, along with other cofounding factors, like strength deficits of the hip abductors the socket [46]. The strength status of people with TFA is therefore of high clinical relevance.”

This recommendation does remove “So, in socket design it may be advisable to grantee passive stability and a very good force transmission, while not restricting the remaining muscles.” however, the proposed change does capture the importance of socket design.

Response to the reviewer: We would like toWe took over most of the suggestions of the reviewer.

499: “K_level” to “K-level”

Response to the reviewer: Changes

Conclusion:

Looks good!

Response to the reviewer: Thank you!

Reviewer #3: The authors made significant improvements to the manuscript. Overall, the writing is better and the authors have addressed my and other reviewer's comments. There a few minor issues to revise before being ready for publication.

Response to the reviewer: We thank the reviewer for these kind words. We also acknowledge the efforts und time of the reviewer. Thank you!

Slight formatting issues with table 1, column width is too narrow.

Response to the reviewer: We adapted this. We guess that such issues will be taken care of by the editorial office and the formatting of the tables will be adjusted to fit to their layout.

Line 28: I am assuming you mean mass, not weight.

Response to the reviewer: We apologize, but we do not know exactly which line the reviewer was referring to. We searched the document and changed weight to mass.

Line 443: grantee, incorrect word?

Response to the reviewer: Apologies for the typo, this supposed to say “guarantee”. This section was shortened and rewritten and this part was deleted.

Repeated use of consequentially, a different word or phrase in few instances to improve readability

Response to the reviewer: Yes, we seem to like this word too much. We changed the manuscript accordingly.

Check for consistent spacing after periods, fluctuates between 0 to 2 spaces

Response to the reviewer: We checked spacing.

Check - versus _ in regards to K-levels (e.g., lines 498-499)

Response to the reviewer: Changed

Thank you for addressing our comments.

Response to the reviewer: Your comments are appreciated and help to improve the manuscript considerably.

---

## [Decision Letter · Decision Letter 2]

28 Jul 2020

PONE-D-19-24706R2

The influence of hip muscle strength on gait in individuals with a unilateral transfemoral amputation

PLOS ONE

Dear Dr. Heitzmann,

Thank you for submitting your manuscript to PLOS ONE. After careful consideration, we feel that it has merit but does not fully meet PLOS ONE’s publication criteria as it currently stands. Therefore, we invite you to submit a revised version of the manuscript that addresses the points raised during the review process.

We look forward to receiving your revised manuscript.

Kind regards,

Yih-Kuen Jan, PhD

Academic Editor

PLOS ONE

Reviewers' comments:

Reviewer's Responses to Questions

**Comments to the Author**

1. If the authors have adequately addressed your comments raised in a previous round of review and you feel that this manuscript is now acceptable for publication, you may indicate that here to bypass the “Comments to the Author” section, enter your conflict of interest statement in the “Confidential to Editor” section, and submit your "Accept" recommendation.

Reviewer #1: All comments have been addressed

Reviewer #2: All comments have been addressed

Reviewer #3: All comments have been addressed

2. Is the manuscript technically sound, and do the data support the conclusions?

Reviewer #1: Yes

Reviewer #2: Yes

Reviewer #3: Yes

3. Has the statistical analysis been performed appropriately and rigorously? 

Reviewer #1: Yes

Reviewer #2: Yes

Reviewer #3: Yes

4. Have the authors made all data underlying the findings in their manuscript fully available?

Reviewer #1: Yes

Reviewer #2: Yes

Reviewer #3: Yes

5. Is the manuscript presented in an intelligible fashion and written in standard English?

Reviewer #1: Yes

Reviewer #2: Yes

Reviewer #3: Yes

6. Review Comments to the Author

Reviewer #1: (No Response)

Reviewer #2: I would like to commend the authors on re-writing their manuscript. This manuscript represents a significant knowledge contribution in terms of how strength deficits can affect walking performance for persons with transfemoral amputation. Below I have listed some minor comments regarding word choice and punctuation.

Line 67: “later” to “lateral”

Line 89: comma between “imaging” and “involved”

Line 90-91: suggest to update the sentence to: “This led to altered walking biomechanics and residual limb muscle activation patterns after a TTA [6]”

Line 118: “Beside” to “Besides”

Line 137: comma between “secondly” and “we”

Line 139: Replace semicolon with period and comma between “Finally” and “we”

line 147: comma between “data” and “maximum”

line 192: comma between “documentation” and “none”

line 220: consider updating the start of this sentence to: “As shown by Pamies-Vila et al., changes of the…”

line 242: comma between “measurements” and “a”

Discussion:

The flow of the discussion has improved greatly! Thank you for taking time and updating this section.

Line 378: “will” to “would”

Line 378: comma between “gait” and “this”

line 386 “lead” to “led”

line 392: “conclude” to “concluded”

line 403: “transition” to “transmission”

Line 403-404: recommend updating the wording of this sentence to: “A reduced power transmission between the stump and the socket may cause the observed muscle weakness to manifest a more pronounced functional deficit.”

Line 406: “transition” to “transmission”

Line 412: comma between “discrepancy” and “was”

Reviewer #3: The authors addressed reviewers comments during this second review, so thank for continued improvements on this manuscript. Readability and clarity have been improved. There are a few more minor changes to be made:

Abstract:

Line 64: "i.e. later[al] trunk"

Line 68: "between later[al]"

Introduction:

Line 92: "ultrasound imaging[,] involved"

Line 149: "movement; [f]inally we"

Methods:

Looks good, no suggested changes.

Results:

Looks good, no suggested changes.

Discussion:

Line 357: "First and foremost" is colloquial, could say "Our data shows that..."

Line 443: "it was conclud[ed]..."

Line 464: Remove 'A' at start of "A decreased..."

Line 465: Remove comma after "factors"

Line 466: Add comma after "discrepancy"

Conclusion:

Looks good, no suggested changes.

Thank you again for your continued work on this manuscript!

7. PLOS authors have the option to publish the peer review history of their article (what does this mean?). If published, this will include your full peer review and any attached files.

Reviewer #1: **Yes: **Scott A. Beardsley

Reviewer #2: No

Reviewer #3: No

---

## [Author Response · Author response to Decision Letter 2]

28 Jul 2020

Reviewer #1: (No Response)

Response from the Authors: We would like to thank the reviewer once again for the input. It helped considerably to increase the quality of the manuscript.

Reviewer #2: I would like to commend the authors on re-writing their manuscript. This manuscript represents a significant knowledge contribution in terms of how strength deficits can affect walking performance for persons with transfemoral amputation. Below I have listed some minor comments regarding word choice and punctuation.

Response from the Authors: We would like to thank reviewer for the positive words.

Line 67: “later” to “lateral”

Response from the Authors: Changed

Line 89: comma between “imaging” and “involved”

Response from the Authors: Changed

Line 90-91: suggest to update the sentence to: “This led to altered walking biomechanics and residual limb muscle activation patterns after a TTA [6]”

Response from the Authors: Changed

Line 118: “Beside” to “Besides”

Response from the Authors: Changed

Line 137: comma between “secondly” and “we”

Response from the Authors: We added a comma

Line 139: Replace semicolon with period and comma between “Finally” and “we”

Response from the Authors: Changed

line 147: comma between “data” and “maximum”

Response from the Authors: We added a comma

line 192: comma between “documentation” and “none”

Response from the Authors: We added a comma

line 220: consider updating the start of this sentence to: “As shown by Pamies-Vila et al., changes of the…”

Response from the Authors: We changed the start accordingly

line 242: comma between “measurements” and “a”

Response from the Authors: We added a comma

Discussion:

The flow of the discussion has improved greatly! Thank you for taking time and updating this section.

Response from the Authors: We would like to thank the reviewer for these positive words. This was also an effort of the reviewer. Thank you for helping to improve the manuscript. 

Line 378: “will” to “would”

Response from the Authors: Changed

Line 378: comma between “gait” and “this”

Response from the Authors: We added a comma

line 386 “lead” to “led”

Response from the Authors: Changed

line 392: “conclude” to “concluded”

Response from the Authors: Changed

line 403: “transition” to “transmission”

Response from the Authors: Changed, we further updated the sentence as suggested.

Line 403-404: recommend updating the wording of this sentence to: “A reduced power transmission between the stump and the socket may cause the observed muscle weakness to manifest a more pronounced functional deficit.”

Response from the Authors: Thank you for this suggestion, we updated the sentence as suggested.

Line 406: “transition” to “transmission”

Response from the Authors: Changed

Line 412: comma between “discrepancy” and “was”

Reviewer #3: The authors addressed reviewers comments during this second review, so thank for continued improvements on this manuscript. Readability and clarity have been improved. There are a few more minor changes to be made:

Response from the Authors: We also thank the reviewers for their time and effort. Without their help we would not come to this stage of the manuscript.

Abstract:

Line 64: "i.e. later[al] trunk"

Line 68: "between later[al]"

Response from the Authors: Changed. Apologies for these typos

Introduction:

Line 92: "ultrasound imaging[,] involved"

Response from the Authors: Changed

Line 149: "movement; [f]inally we"

Response from the Authors: We changed this sentence in accordance to the suggestion of reviewer 2. 

Methods:

Looks good, no suggested changes.

Response from the Authors: Thank you.

Results:

Looks good, no suggested changes.

Response from the Authors: Thank you.

Discussion:

Line 357: "First and foremost" is colloquial, could say "Our data shows that..."

Response from the Authors: We would like to thank the reviewer for that suggestion and changed the sentence accordingly.

Line 443: "it was conclud[ed]..."

Response from the Authors: Changed

Line 464: Remove 'A' at start of "A decreased..."

Response from the Authors: Changed

Line 465: Remove comma after "factors"

Response from the Authors: Deleted

Line 466: Add comma after "discrepancy"

Response from the Authors: Added

Conclusion:

Looks good, no suggested changes.

Response from the Authors: Thank you!

Thank you again for your continued work on this manuscript!

Response from the Authors: It was our pleasure. Thanks

---

## [Decision Letter · Decision Letter 3]

11 Aug 2020

The influence of hip muscle strength on gait in individuals with a unilateral transfemoral amputation

PONE-D-19-24706R3

Dear Dr. Heitzmann,

We’re pleased to inform you that your manuscript has been judged scientifically suitable for publication and will be formally accepted for publication once it meets all outstanding technical requirements.

Kind regards,

Yih-Kuen Jan, PhD

Academic Editor

PLOS ONE

Additional Editor Comments (optional):

Reviewers' comments:

Reviewer's Responses to Questions

**Comments to the Author**

1. If the authors have adequately addressed your comments raised in a previous round of review and you feel that this manuscript is now acceptable for publication, you may indicate that here to bypass the “Comments to the Author” section, enter your conflict of interest statement in the “Confidential to Editor” section, and submit your "Accept" recommendation.

Reviewer #1: All comments have been addressed

Reviewer #2: All comments have been addressed

Reviewer #3: All comments have been addressed

2. Is the manuscript technically sound, and do the data support the conclusions?

Reviewer #1: Yes

Reviewer #2: Yes

Reviewer #3: Yes

3. Has the statistical analysis been performed appropriately and rigorously? 

Reviewer #1: Yes

Reviewer #2: Yes

Reviewer #3: Yes

4. Have the authors made all data underlying the findings in their manuscript fully available?

Reviewer #1: Yes

Reviewer #2: Yes

Reviewer #3: Yes

5. Is the manuscript presented in an intelligible fashion and written in standard English?

Reviewer #1: Yes

Reviewer #2: Yes

Reviewer #3: Yes

6. Review Comments to the Author

Reviewer #1: (No Response)

Reviewer #2: I would like to thank the authors once again for their continued work through this review process. The authors have a well written and scientifically sound manuscript. I have included a few minor suggestions. I feel that this manuscript is ready for publication.

Make sure that there is consistent punctuation after e.g. and i.e. Sometimes the authors include a comma after these abbreviations and sometimes not.

Line 220: comma after “al.”

Line 221: “effects” to “effect”, “in” to “on”

Line 371: remove “was”

Line 404: there is an extra period

Reviewer #3: The authors have made the necessary changes to the manuscript and now have an article that is ready for publication. Thank you for your efforts!

7. PLOS authors have the option to publish the peer review history of their article (what does this mean?). If published, this will include your full peer review and any attached files.

Reviewer #1: **Yes: **Scott A. Beardsley

Reviewer #2: No

Reviewer #3: No

---

## [Editor Report · Acceptance letter]

25 Aug 2020

PONE-D-19-24706R3 

The influence of hip muscle strength on gait in individuals with a unilateral transfemoral amputation 

Dear Dr. Heitzmann:

I'm pleased to inform you that your manuscript has been deemed suitable for publication in PLOS ONE. Congratulations! Your manuscript is now with our production department. 

Kind regards, 

on behalf of

Dr. Yih-Kuen Jan 

Academic Editor

PLOS ONE